# GENESIS CGDYN: large-scale coarse-grained MD simulation with dynamic load balancing for heterogeneous biomolecular systems

Jaewoon Jung [1,2,4], Cheng Tan [1,4] & Yuji Sugita [1,2,3] ✉

Residue-level coarse-grained (CG) molecular dynamics (MD) simulation is widely used to investigate slow biological processes that involve multiple proteins, nucleic acids, and their complexes. Biomolecules in a large simulation system are distributed non-uniformly, limiting computational efficiency with conventional methods. Here, we develop a hierarchical domain decomposition scheme with dynamic load balancing for heterogeneous biomolecular systems to keep computational efficiency even after drastic changes in particle distribution. These schemes are applied to the dynamics of intrinsically disordered protein (IDP) droplets. During the fusion of two droplets, we find that the changes in droplet shape correlate with the mixing of IDP chains. Additionally, we simulate large systems with multiple IDP droplets, achieving simulation sizes comparable to those observed in microscopy. In our MD simulations, we directly observe Ostwald ripening, a phenomenon where small droplets dissolve and their molecules redeposit into larger droplets. These methods have been implemented in CGDYN of the GENESIS software, offering a tool for investigating mesoscopic biological processes using the residue-level CG models.

Computational simulations with modeling of biomolecular structure and dynamics at various levels of detail can elucidate complex cellular phenomena in close collaboration with experimental studies. Quantum mechanics/molecular mechanics (QM/MM) or atomistic molecular dynamics (MD) methods provide detailed descriptions of the conformational dynamics of target molecular systems. However, they are computationally demanding when exploring long-time dynamics of a large biomolecule or a biomolecular system consisting of many biomolecules. Coarse-grained (CG) MD simplifies the description of these systems by representing multiple atoms as a single particle and thereby reduces computational complexity[1–3]. CG MD simulations retain essential structural and dynamic properties in biomolecules[2], making them valuable for investigating biomolecular processes, such as protein folding and dynamics[4–6], protein-DNA interactions[7,8], nucleosome dynamics[9,10], genome organizations[11,12],

condensate formation/destruction via liquid-liquid phase separation (LLPS)[13–17], etc. Among many CG methods, residue-level CG MD simulations for proteins, nucleic acids, and lipids can bridge the gap between atomistic simulations and experimental observations[2]. They provide insights into fundamental but complex biological processes by balancing modeling accuracy and computational efficiency.

Various CG models have been developed to capture the structure, dynamics, and inter-molecular interactions of biomolecules. These models include the structure-based Gō model[4] and its variants for folded biomolecules (such as AICG2+ [5]), the HPS model for intrinsically disordered proteins (IDPs)[14], the 3SPN series models for nucleic acids[18–20], and the Martini[21] and SPICA[22] models for lipid systems. Other notable CG models, for instance, UNRES[23], OPEP[24], and PRIMO/PRIMONA[25], have been specifically designed to address different aspects of biological phenomena. Several MD programs, including

[1]Computational Biophysics Research Team, RIKEN Center for Computational Science, Kobe, Hyogo 650-0047, Japan. [2]Theoretical Molecular Science Laboratory, RIKEN Cluster for Pioneering Research, Wako, Saitama 351-0198, Japan. [3]Laboratory for Biomolecular Function Simulation, RIKEN Center for Biosystems Dynamics Research, Kobe, Hyogo 650-0047, Japan. [4]These authors contributed equally: Jaewoon Jung, Cheng Tan. ✉e-mail: sugita@riken.jp

CHARMM[26], GROMACS[27], OpenMM[28], NAMD[29], HOOMD-blue[30], LAMMPS[31], Cafemol[32], and GENESIS[33–35], offer a diverse set of tools and capabilities for performing CG MD simulations in various contexts. CG models are categorized into two: with explicit solvent molecules and with implicit solvent approximation. For the CG models with implicit solvent, while they have proven invaluable in addressing biological problems, there are difficulties in their implementations that do not arise in atomistic MD simulations. When CG MD simulation with the implicit solvent approximation is parallelized on multiple processors by the conventional domain decomposition scheme with an equal domain size for all processes, the processes in charge of dense regions have a significant workload. In contrast, those for dilute (or sparse) regions are almost idle, leading to non-negligible waiting time to synchronize tasks between all the processes. Moreover, the incorporation of diverse potential functions that describe interactions between different biomolecular components adds to the complexity of optimizing CG MD software[35].

These difficulties limit the available system size even for residue-level CG models with the implicit solvent approximations. This requires innovative CG MD simulation schemes. Scientifically, on the other side, there is a growing demand for residue-level CG MD simulations of large biological systems. For instance, protein/nucleic acid condensates (or droplets) formed by LLPS have attracted many chemists and biologists due to their relevance to serious neurotoxic diseases or essential biological functions in the cellular cytoplasm or nucleus[13,14,36,37]. Currently, standard simulations of LLPS have used so-called slab models[14], which have two short-length dimensions and only one long dimension within a periodic rectangular box. Dense and dilute phases are observed along the long dimension in equilibrium conditions. However, this anisotropic shape may not fully capture the three-dimensional (3D) nature of LLPS. Another example is the 3D modeling of chromatin. Many computational models of 3D structures of chromatin have been proposed using experimental data such as Hi-C[38,39]. However, due to the dynamic nature of chromatin structures and computational limitations, most of those structural models are developed at relatively low resolutions (kilo-bases)[38,39]. On the other side, at higher resolution, conformational dynamics of only a small number of nucleosomes with/without transcription factors were simulated with residue-level CG models[9,10].

To perform CG MD simulations of non-uniform densities, dynamic load balancing is essential to accommodate rapid but significant changes in particle distributions in biological processes such as droplet formations from evenly distributed proteins. Various endeavors have been undertaken to enhance the efficiency of MD simulations through the development of dynamic load balancing schemes. LAMMPS employs the recursive coordinate bisection (RCB) algorithm for dynamic load balancing[31]. In GROMACS, domain sizes are dynamically adjusted based on computational time for each process[40]. The ddcmd program introduces domain decomposition based on Voronoi cells[41]. The ESPResSo software utilizes domain decomposition based on space-filling curve, with dynamic re-balancing achieved through a collection of adaptive octrees[42]. Guzman et al. proposed a domain decomposition scheme suitable for multi-scale simulations by assigning different domain sizes for all-atom and CG models[43]. Additionally, Grime and Voth developed a highly scalable scheme for ultra-coarse-grained models using the Hilbert space-filling curve[44]. Some have employed the kd-tree schemes for dynamic load balancing[45,46].

In this work, we have developed a unique domain decomposition scheme with dynamic load balancing to enable efficient residue-level CG MD simulations on parallel computers to handle non-uniform densities in a large biological system. We have implemented it in the GENESIS software[33–35] as an MD engine called CGDYN (CG molecular DYNamics), specifically designed for CG MD simulations. CGDYN is optimized for parallel computers with many processors based on a domain decomposition with load balancing scheme akin to LAMMPS

and programs utilizing kd-tree algorithms. In addition, we implement a united neighboring list search algorithm to calculate the energy and forces of diverse potential functions with different cutoff values in residue-level CG MD simulations, and thereby CGDYN outperforms other MD programs in terms of computational performance. In this study, we utilize CGDYN to investigate molecular mechanisms underlying the fusion of two smaller droplets into a larger one using AICG2+[5] and HPS[14] models. Additionally, we conduct ultra-large-scale CG MD simulations consisting of multiple droplets to observe droplet formations, whose sizes are almost equivalent to the confocal microscope images. CGDYN in GENESIS provides a computational tool for investigating mesoscale biological phenomena at the residue-level descriptions and connecting our understanding of the structure and dynamics of proteins and nucleic acids with cellular-scale biological phenomena.

## Results

### Domain decomposition scheme in CGDYN

We have developed a unique domain decomposition scheme with dynamic load balancing to parallelize the residue-level CG MD simulations. The scheme is based on the midpoint cell method[47] used in GENESIS SPDYN for atomistic MD simulations. The midpoint cell method divides a simulation space hierarchically: subdomains at first and then small cells from each subdomain. The cell size is decided from the interaction-range threshold, and the number of subdomains equals the number of processes. Because of almost uniform particle distributions in atomistic MD simulations, every subdomain has the same number of cells in SPDYN[33]. In contrast, each subdomain in CGDYN includes a different number of cells to avoid load imbalances by adopting a domain decomposition scheme which we call the cell-based kd-tree method. Here, subdomains for low particle density regions include more cells, while those for high-density regions contain fewer cells.

How to assign cells to each subdomain is described in Fig. 1a. We first divide a simulation system into two subdomains by making a boundary of cells such that two subdomains have nearly the same number of particles. Each subdomain is again divided into the next-level subdomains, in which their numbers of particles are almost identical. We iterate this procedure until the number of subdomains becomes the same as the process numbers. Each subdomain has particle data from the cell in the subdomain and the adjacent cells in other ones to compute bonded and nonbonded interactions (Fig. 1b). To complete this data structure, communication between subdomains, namely, sending particle coordinates from the boundary cells in one subdomain (i.e., the rank 5 in Fig. 1b) to the neighbors (the ranks 1, 4, 6, and 9) and receiving the coordinates from the boundary cells in the neighbors (the ranks 1, 4, 6, and 9) to the target subdomain (the rank 5), are required for the energy/force evaluation. In the residue-level CG MD simulations, particle densities in subdomains and cells can be changed rapidly. In such cases, the initial domain decomposition based on the cell-based kd-tree method cannot guarantee a good load balance. As an example, we show a simulation of the Heat-resistant obscure protein-11 (Hero-11)[48] and the low-complexity domain of TDP-43 (TDP-43-LCD, with amino acid residues 261-414)[49]. TDP-43-LCD is known to form droplets in physiological conditions, while highly-charged Hero-11 can regulate the TDP-43-LCD condensate[48,50]. At $t = 0$, the particle densities of the subdomains that include the TDP-43-LCD droplet are quite high, while the rest include just Hero-11 proteins at low concentrations. As the simulation progresses, the particle densities become more uniform, requiring different domain decompositions from the initial one (Fig. 1c). During the simulation, we decompose the simulation space using the cell-based kd-tree method at a fixed interval (around $10^6$ steps but it depends on the integration time step and the target system). This example suggests the importance of dynamic load balancing to keep the best performance of long-time CG MD simulations. Our load balance scheme is similar to the RCB

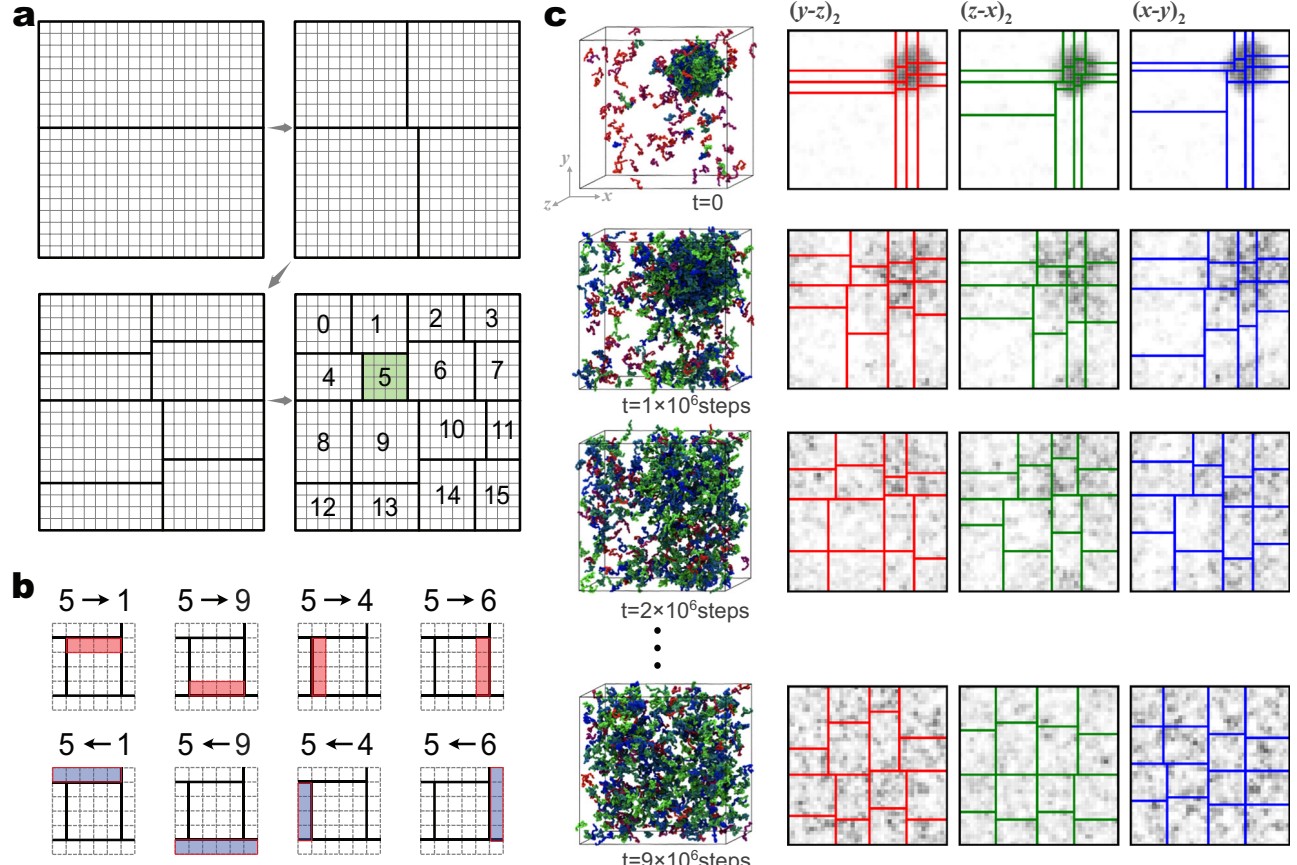

**Fig. 1 | Domain decomposition scheme in CGDYN. a** The domain decomposition algorithm with the cell-based kd-tree method. **b** Communication in sending the coordinates from MPI rank 5 to other processes (upper) and receiving the coordinates from other processes to MPI rank 5 (lower). **c** Updates of subdomains during MD simulations of Hero11 and TDP-43-LCD. The first column displays snapshots of molecular structures at specific MD steps. The remaining three columns on the right illustrate the expansion of subdomains across three dimensions and on the second layer of a $4 \times 4 \times 4 = 64$ domain decomposition. Source data of (**c**) are provided as a Source Data file.

algorithm implemented in LAMMPS[31], ls1 mardyn[46], and other MD programs using a kd-tree algorithm. However, CGDYN is distinguishable from them by incorporating more complicated potential functions with multiple cutoff distances.

## CGDYN structure

In CGDYN, the cell size is greater than or equal to half of the distance wherein the neighbor list for electrostatic interaction is considered. All particle information in each subdomain (coordinate, force, charge, atom class number, and so on) is saved cell-wise (Supplementary Fig. 1). We make an identifier of particles (ordered/disorder protein region, DNA base, and so on) for efficient neighbor list generation. For each cell, we first locate the information on charged particles followed by uncharged particles in this order. The array of potential function types and parameters of bonded interactions (bond/angle/dihedral angle list) is prepared without using cell indices (Supplementary Fig. 1). In the case of bond, the program writes parameters of quadratic and next quartic bond terms sequentially in bond-related array. There are three potential functions for angle terms:

$$
\begin{aligned}
E_{\text{angle}}^{(1)} &= \sum_{\theta_i \in \text{angles}} -k_B T \ln \frac{P_\theta(\theta_i | i)}{\sin \theta_i} \\
E_{\text{angle}}^{(2)} &= \sum_{r_i \in 1-3 \text{pairs}} -\varepsilon_i \exp\left(\frac{-(r_i - r_{i,0})^2}{2w_i^2}\right) \\
E_{\text{angle}}^{(3)} &= \sum_{\theta_i \in \text{angles}} k_{a,i} (\theta_i - \theta_{i,0})^2 .
\end{aligned}
\tag{1}
$$

The program first writes parameters of $E_{\text{angle}}^{(1)}$ and next writes parameters $E_{\text{angle}}^{(2)}$ and $E_{\text{angle}}^{(3)}$. Similarly, we have three dihedral angle potential functions:

$$
\begin{aligned}
E_{\text{dihe}}^{(1)} &= \sum_{\varphi_i \in \text{dihedrals}} -k_B T \ln P_d(\varphi_i | i) \\
E_{\text{dihe}}^{(2)} &= \sum_{\varphi_i \in \text{dihedrals}} -\epsilon_{\varphi,i} \exp\left(\frac{-(\varphi_i - \varphi_{i,0})^2}{2\sigma_{\varphi,i}^2}\right) \\
E_{\text{dihe}}^{(3)} &= \sum_{\theta_i \in \text{dihedrals}} \sum_n k_{\phi,i,n} \left[1 + \cos\left(n(\phi_i - \phi_{i,0})\right)\right],
\end{aligned}
\tag{2}
$$

and the program writes parameters sequentially in dihedral angle array.

All nonbonded interactions, including electrostatic, HPS, excluded volume, etc, have different interaction ranges. Among them, electrostatic interaction has the longest interaction range. For a given particle, the interaction range and the cell involved in the calculation are depicted in Supplementary Fig. 2. For bonded and excluded volume interactions, we consider the particles in the target cell and neighboring cells. Electrostatic and HPS potential functions have longer interaction ranges. Therefore, particles in the next neighboring cells from the target cell are considered.

We generate the neighbor lists concurrently for all nonbonded interactions except for electrostatic and protein-DNA terms (Supplementary Algorithm). First, we predefine threshold values of the neighbor list generation for excluded volume, DNA pairing, and HPS, which are denoted here as $r_{\text{p,exv}}$, $r_{\text{p,dna}}$, $r_{\text{p,hps}}$, respectively. If a pairwise

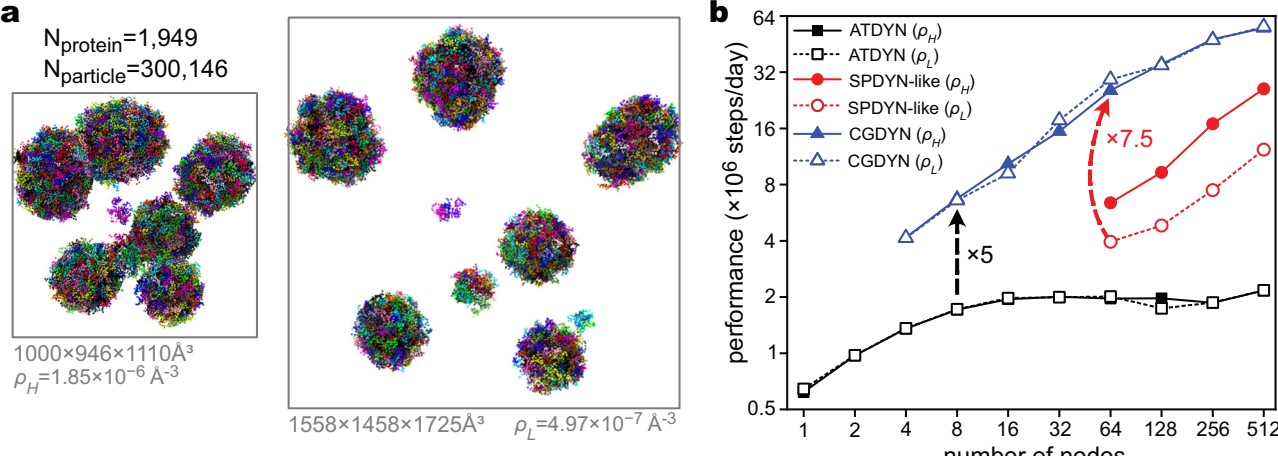

**Fig. 2 | Benchmark results of CG MD simulations with three algorithms: ATDYN, SPDYN-like and CGDYN. a** The two benchmark systems with different densities. **b** Benchmark performance ($\times 10^6$ steps/day). ATDYN can be used with a small number of nodes, but the parallel efficiency is low, and the performance is saturated from 16 nodes. The SPDYN-like algorithm performs better using many nodes but has a higher dependency on particle density. CGDYN performs better than MD simulations based on the ATDYN and SPDYN-like algorithms, showing a weak dependency on particle densities. Source data are provided as a Source Data file.

distance between two particles is shorter than $r_{p,exv}$, we consider the particles in the neighbor lists for all the nonbonded interactions. If the distance, $r$, is in the range of $r_{p,exv} < r < r_{p,dna}$, we do not consider the particles in the neighboring list of excluded volume. If the distance exceeds $r_{p,dna}$, the particles are included in the neighbor lists of the HPS interactions only. The neighbor lists of electrostatic interaction are generated separately by evaluating pairwise distances only between charged particles with predefined threshold value, $r_{p,ele}$. Threshold values of these interactions in the neighbor search are greater than those in the nonbonded energy/force calculations with $r_{p,XX} = r_{c,XX} + r_{buffer}$ ($r_{p,XX}$ and $r_{c,XX}$ are threshold of neighbor list generation and energy/force calculations for XX interactions, respectively). Neighbor lists for PWMCos and sequence-nonspecific hydrogen bond (HB) potential used in protein-DNA interactions are also considered separately. The frequency of neighbor list search is user-defined value, but the program skips the neighbor search if the maximum particle displacement is less than half of $r_{buffer}$.

To accelerate the evaluations of nonbonded interactions, SIMD (Single Instruction, Multiple Data) is applied to electrostatic and HPS force calculations by doing calculations even for unnecessary pairs with the pairwise distance between $r_{c,XX}$ and $r_{p,XX}$. For other interactions including the excluded volume, DNA base pairing, SIMD is not applied because the ratio of $\frac{r_{p,XX}^3}{r_{c,XX}^3}$ is much larger than 1 and it is better to skip unnecessary calculations for the pairwise distance range between $r_{c,XX}$ and $r_{p,XX}$ using conditional statements instead of applying SIMD.

Dynamic load balancing with the cell-based kd-tree scheme is applied at fixed intervals (which we will name the load balance update period), defined by application users. In this procedure, we do not change the cell size and only change the cell assignment to each subdomain. This process is applied as a default when we start MD. If the MD simulation time is the multiple of the load balance update period, the cell reassignment to each subdomain is done in a procedure like Fig. 1a. Communication between neighboring processes is redefined in a procedure like Fig. 1b. The subdomain information is saved in the global data array and the global data is completed by collective communication. Each subdomain reads subdomain information from the global data and MD simulations continue based on the new subdomain data (Supplementary Fig. 3).

## Benchmark tests of CGDYN for heterogeneous biological systems

We examined the computational performance of CGDYN in CG MD simulations of heterogeneous multiple droplet systems. Two multiple-droplet systems with different particle densities were prepared for the performance comparison: one with $\rho = 1.85 \times 10^{-6}$ and the other with $\rho = 4.97 \times 10^{-7}$ (chains per $\text{Å}^3$, Fig. 2a). Both systems consist of 1949 TDP-43-LCD chains (number of particles is 300,146). We compared the computational performances for three different algorithms in GENESIS: (i) CGDYN (the cell-based kd-tree method with dynamic load balancing), (ii) SPDYN-like (the original midpoint cell method without dynamic load balancing), and (iii) ATDYN (atomic decomposition without dynamic load balancing). ATDYN shows no performance dependence on the two systems, but the computational efficiency is limited to $2.0 \times 10^6$ steps/day (Fig. 2b). In comparison, MD simulations using CGDYN show 3–30 times better performances than ATDYN (Fig. 2b). Importantly, almost identical speeds are obtained between the high and low-density systems with CGDYN due to the efficient parallelization with dynamic load balancing, suggesting that these algorithms used in CGDYN work well irrespective to the particle densities. MD simulations with CGDYN accelerate at most 7.5 times compared to the SPDYN-like algorithms (Fig. 2b). The acceleration with CGDYN can depend on the frequency of the dynamic load balancing. Supplementary Fig. 4 shows that the twice better performance is observed by applying the dynamic load balancing 100 times more frequently.

We found that CGDYN works well for very large systems with more than 2.5 million CG particles (Supplementary Fig. 5). It shows good scalability up to 4096 nodes on Fugaku (16,384 MPI processes) with ~$5.0 \times 10^7$ steps/day, promising to simulate very large biological systems efficiently with residue-level CG models. We also conducted benchmark tests on systems previously published in our work[35]. CGDYN encountered memory issues on the RIKEN supercomputer Hokusai when using a small number of processes for large systems. Nevertheless, we managed to execute MD simulations even for large systems by increasing the number of processes, resulting in CGDYN outperforming ATDYN across all systems with a ten-fold speedup (Supplementary Fig. 6).

To evaluate the efficiency of CG MD simulations using CGDYN, it is essential to compare it with other MD programs. Supplementary

Fig. 7 shows the performance comparison of residue-level CG MD simulations of DNA systems among ATDYN, CGDYN, and Open3SPN2[51]. Although Open3SPN2 is based on OpenMM[28] and accelerated with GPU processors, CGDYN on a single node performs better on multiple double-stranded DNA systems than Open3SPN2. For larger DNA systems, the superiority of computational performance with CGDYN is more significant, promising efficient residue-level CG MD simulations of mesoscopic biological systems (Supplementary Table 1).

The comparison between CGDYN and Open3SPN2 regarding dynamic load balancing effects is not straightforward. Hence, we assessed the performance of GENESIS and GROMACS on Fugaku by creating clusters of DPPC micelles using the dry Martini force field[52]. Supplementary Fig. 8 reveals that GROMACS exhibits similar performance to CGDYN for 32 nodes. However, CGDYN surpasses GROMACS from 64 nodes onwards, and this performance margin widens with an increasing number of processes. Our primary aim in comparing the performance of the dry Martini systems is not to demonstrate the superiority of CGDYN within the dry Martini model but to ascertain whether its parallelization is comparable to existing MD software equipped with robust dynamic load balancing schemes. We also observed that the effect of dynamic load balancing within the dry Martini model is not as pronounced as that of the residue-level CG model. This is primarily attributed to the small cutoff distance in nonbonded interactions within the dry Martini model, resulting in a lower computation-to-communication ratio. We additionally evaluated the performance for a smaller system (150,000 particles), wherein GROMACS exhibited superior performance to CGDYN from 16 to 128 nodes. Considering this, GROMACS appears to be more optimized than CGDYN for energy/force calculation, whereas CGDYN demonstrates superior scalability for larger systems with nonuniform particle densities.

Furthermore, we compared CGDYN with LAMMPS using the HPS potential outlined in Eq. (7) in the Methods section. Both CGDYN and LAMMPS exhibit saturated performance beyond 512 nodes. Across all process counts, CGDYN outperforms LAMMPS. Although LAMMPS approaches similar performance levels as CGDYN with an increasing number of processes, even at 1024 nodes, CGDYN demonstrates 1.3 times better performance than LAMMPS, with a reduced amount of communication time (CGDYN: 0.5 ms/step and LAMMPS: 0.75 ms/step). The superior scalability of LAMMPS compared to CGDYN is mainly attributed to a larger computation-to-communication ratio.

## Molecular mechanisms for the fusion of two droplets

The fusion of phase-separated liquid-like droplets is an important mechanism for maintaining stable cellular environments when the concentration of components changes[53]. The residue-level CG MD simulations[13,14,37] allow the investigation of inter-molecular interactions and the resulting diffusion/mixing of different components during the droplet fusions. However, MD simulations of this process are computationally demanding and not frequently employed in practical studies due to the large number of molecules involved and the rapid exchange of components in droplets. In this study, we use CGDYN to examine the fusion of two separated droplets, each consisting of approximately 500 chains of TDP-43-LCD (resulting in 1000 chains in total, Fig. 3a). The HPS model[14] was utilized for the entire protein, except for a short α-helical region (residues 320 to 334) modeled with AICG2+[5].

The MD simulations were conducted at $T = 280K$ for $1 \times 10^8$ steps to explore the dynamics of the system. Despite rapid density changes during the simulations, we achieved a speed of $1.25 \times 10^7$ steps per day on a single node (16 MPI processes in conjunction with 3 OpenMP threads) on a local PC cluster (Intel Xeon Gold 6240 R CPU, 2.4 GHz). The DBSCAN clustering analysis[54] is employed to identify chains belonging to the two distinct droplets in the initial structure (Figs. 3a

and 3b). By labeling each chain, we can track their positions and monitor the fusion process (Fig. 3a–g). Interestingly, the fusion of two droplets occurs shortly after the start of the MD simulation and completes within the first several $10^7$ MD steps (Fig. 3g). To investigate the interaction and redistribution of components within the droplets during fusion, we monitored the "mixing" of contents and the associated shape changes.

Through structural analysis and calculation of chain distribution in the simulation box, we observed that the two droplets merge into one at approximately $2 \times 10^7$ steps, although the shape of the merged droplet is a flattened ellipsoid, and the chains from the original droplets are not yet fully mixed (Fig. 3c, d). After $1 \times 10^8$ steps of simulations, the chains from both droplets mixed completely (Fig. 3e, f). The entire fusion process can be viewed in Supplementary Movie 1. To quantify the relationship between the content mixings and the shape changes, we introduce five order parameters. One parameter, $D_{I,J}$, represents the average distances between the center-of-masses (COMs) of chains in the original droplets $I$ and $J$ ($I,J = 1$ or 2, see Methods). $D_{1,1}$ and $D_{2,2}$ increase during the mixing process, while $D_{1,2}$ decreases (Fig. 3h), depicting the redistribution of chains from their original droplet into the fused one. We further define $m$ as $\sqrt{D_{1,1}D_{2,2}}/D_{1,2}$ to incorporate information from $D_{1,1}$, $D_{2,2}$, and $D_{1,2}$. As expected, $m$ increases during the fusion process. Another coordinate, $\eta$, is used to describe the anisotropy of the droplet during fusion (see Methods for definition). Notably, we observe that mixing ($m$) and shape change ($\eta$) are coupled, with the fused droplet reaching a sphere-like shape ($\eta < 4.0$) slightly earlier than the complete mixing of contents ($m \approx 1.0$, Fig. 3i).

## Toward the observation of ultra-large droplets that are detectable by experimental confocal microscopy

In contrast to the droplet formation or regulations studied in MD simulations, the real droplet sizes observed with confocal microscopy vary from sub-micrometer to tens of micrometers[55,56]. To connect a large gap between the simulations and experiments, we focus on the fusion of multiple droplets toward a much larger one, which happens in the cell during the formation of mesoscopic-scale protein droplets due to LLPS[55,57,58]. Here, we generated large systems comprising 16,657 chains of TDP-43-LCD, in total 2,565,178 CG particles. Two systems were created with different overall chain densities: $1.85 \times 10^{-6}Å^{-3}$ (high density) and $6.75 \times 10^{-7}Å^{-3}$ (low density), respectively (Fig. 4a, b). The high-density and low-density systems were simulated in boxes measuring $2087 \times 2067 \times 2077Å^3$ and $2899 \times 2903 \times 2929Å^3$, respectively. Both simulations were performed in the NVT ensemble at $T = 290K$. Leveraging the computational power of the supercomputer Fugaku, we achieved a simulation speed of $2.8 \times 10^7$ steps per day (~280 ns/day, with a time integration step size of 10 fs) on 512 nodes, regardless of the particle densities of the two systems. In total, $1.225 \times 10^9$ and ~$1.200 \times 10^9$ steps were performed for the high- and low-density systems, respectively.

To monitor the dynamics of droplets, we employ the DBSCAN clustering analysis[54] on selected snapshots obtained at different stages during the simulations. In the initial structures, proteins are distributed into more than 50 droplets of varying sizes, ranging from 5 to 452 chains in a droplet. While the two systems exhibit similar droplet size distributions in the initial structures, more drastic changes happen in the high-density system as the simulation progresses. In the high-density system, the number of droplets ($n_d$) quickly decreases from $n_d = 53$ to $n_d = 7$ within the first $5 \times 10^7$ steps, while the largest droplet size increases from $s_d = 452$ at $t = 0$ to $s_d = 6422$ at $t = 5 \times 10^7$ steps (Fig. 4c), indicating the occurrence of droplet fusions. Indeed, a snapshot at $1.00 \times 10^8$ step of the high-density system reveals the formations of a few large droplets of TDP-43-LCD, with the largest droplet exhibiting a non-spherical shape (Fig. 4a). These droplets further assembled into two after ~$3 \times 10^8$ steps. Eventually, after

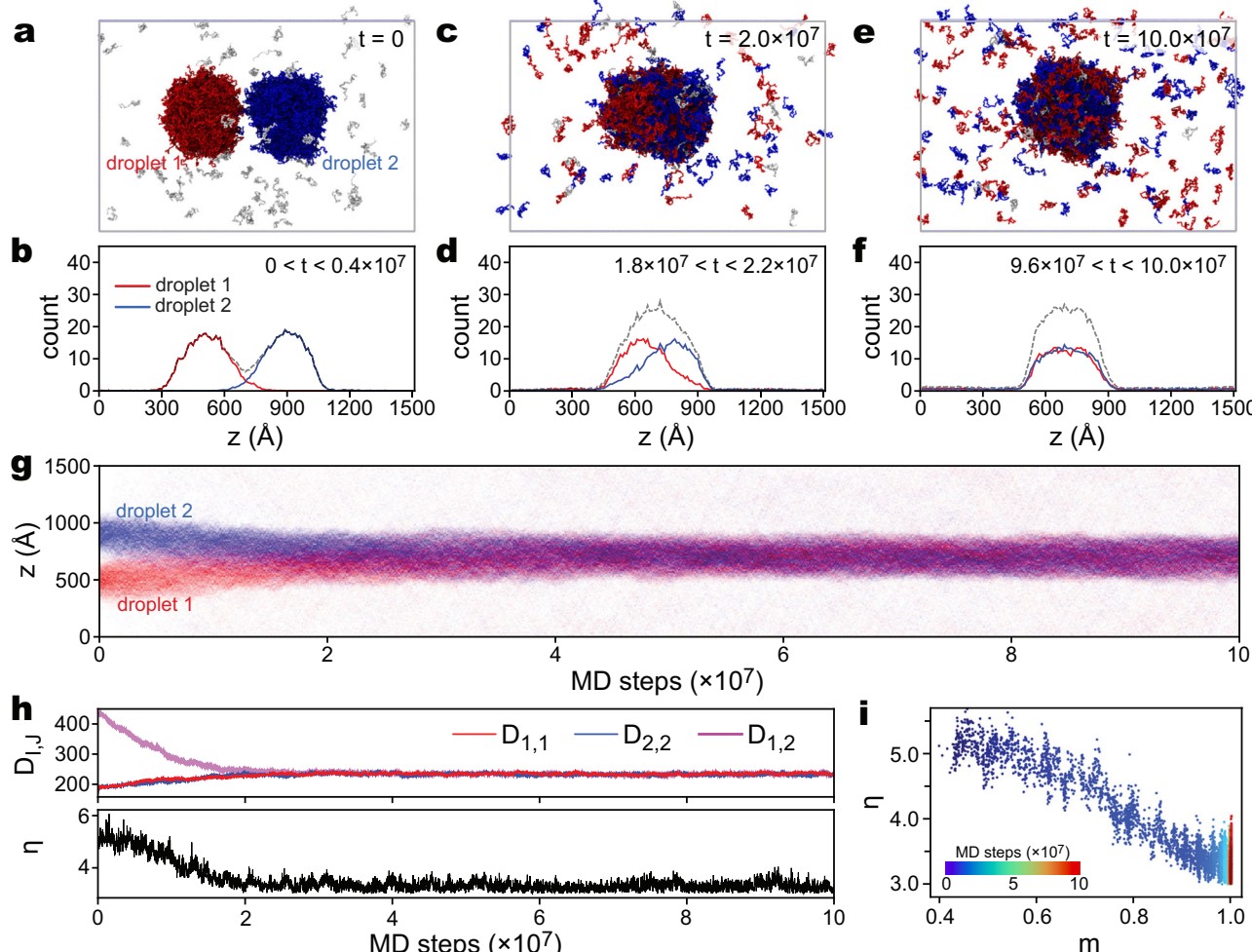

**Fig. 3 | CG MD simulation of the fusion process of two TDP-43-LCD droplets.** The system consists of 1000 chains of TDP-43-LCD. In the initial structure, two separate droplets were put close to each other. The system was simulated at 280 K for $10^8$ steps. Snapshots of simulated structures at $t = 0$ (**a**), $t = 2 \times 10^7$ steps (**c**), and $1 \times 10^8$ steps (**e**). Chains from the two droplets in the initial structure are colored red and blue, respectively. Time-averaged distributions of TDP-43 chains along the z axis during $0 < t < 0.4 \times 10^7$ (**b**), $1.8 \times 10^7 < t < 2.2 \times 10^7$ (**d**), and $9.6 \times 10^7 < t < 10.0 \times 10^7$ (**f**)

steps, respectively. **g** Density of TDP-43 particles along the z axis as a function of simulation time. **h** Time series of average chain-chain distances $D_{I,J}$ (upper) and shape coordinate $\eta$ (lower, $\eta = 3$ indicates perfectly symmetrical sphere and larger $\eta$ values indicate deviations from spherical symmetry). **i** Droplet mixing (depicted by coordinate $m = \sqrt{D_{1,1}D_{2,2}/D_{1,2}}$) against shape change ($\eta$). Detailed definitions of $D_{I,J}$, $\eta$, and $m$ are in the Methods section. Source data are provided as a Source Data file.

$8.5 \times 10^8$ steps, only one droplet remains, comprising ~14,000 chains and a diameter of approximately $0.1 \mu m$ (Fig. 4a, c). In contrast, the number of droplets in the low-density system decreases at a slower pace, reducing from $n_d = 53$ to $n_d = 16$ during the first $4 \times 10^8$ steps and eventually reaching 6 after $1.2 \times 10^9$ steps (Fig. 4c). Simultaneously, the largest droplet size slowly increases from $s_d = 452$ ($t = 0$) to $s_d = 1261$ at $t = 4 \times 10^8$ steps and finally to $s_d = 2,205$ at $t = 1.2 \times 10^9$ steps. The complete procedure of the high-density system is available for viewing in Supplementary Movie 2.

Interestingly, we observed that the decrease in droplet number is not always due to the fusion of smaller droplets into larger ones, as shown in Fig. 3. We discovered that some droplets reduce in size and eventually dissolve into the dilute phase. For instance, in the high-density system, after $4.25 \times 10^8$ steps, the size ($s_d$) of the smaller droplet (depicted as blue in Fig. 4a, middle) decreased from 1297 to 0 (as shown in Fig. 4c, bottom). Meanwhile, larger droplets can grow as the concentration in the dilute phase increases, with more chains entering the dense phase. For example, the size of the larger droplet in the high-density system increased from 12,850 at $t = 4.25 \times 10^8$ to 14,088 at $t = 1.225 \times 10^9$. The differing destinies of large and small droplets can be attributed to the pressure difference sustained across the interface

between the two phases ($\Delta p$), which is described by the Young-Laplace equation, $\Delta p = -2\gamma/R_f$, where $\gamma$ is the surface tension, and $R_f$ is the mean curvature. For small, highly curved droplets, extra energy is required to overcome this large pressure, causing the diffusion of proteins into the dilute phase, and consequently driving Ostwald ripening[59,60].

These simulation results highlight the effectiveness of CGDYN in capturing the fundamental events involved in droplet dynamics with residue-level particle resolutions. We expect that further extensive simulations and in-depth analysis can uncover additional details and refine our understanding of the IDP phase behaviors.

## Discussion
Slab simulations have been commonly employed to investigate the equilibrated phase behavior of biomolecular condensates[13,14,37,49,50]. It is crucial to note that these simulations have limitations in accurately capturing the 3D nature of the fusion processes due to the limited number of chains and the small size of the system. In contrast, the large-scale droplet simulations with the residue-level description of proteins can overcome these potential problems in smaller-scale simulations. As

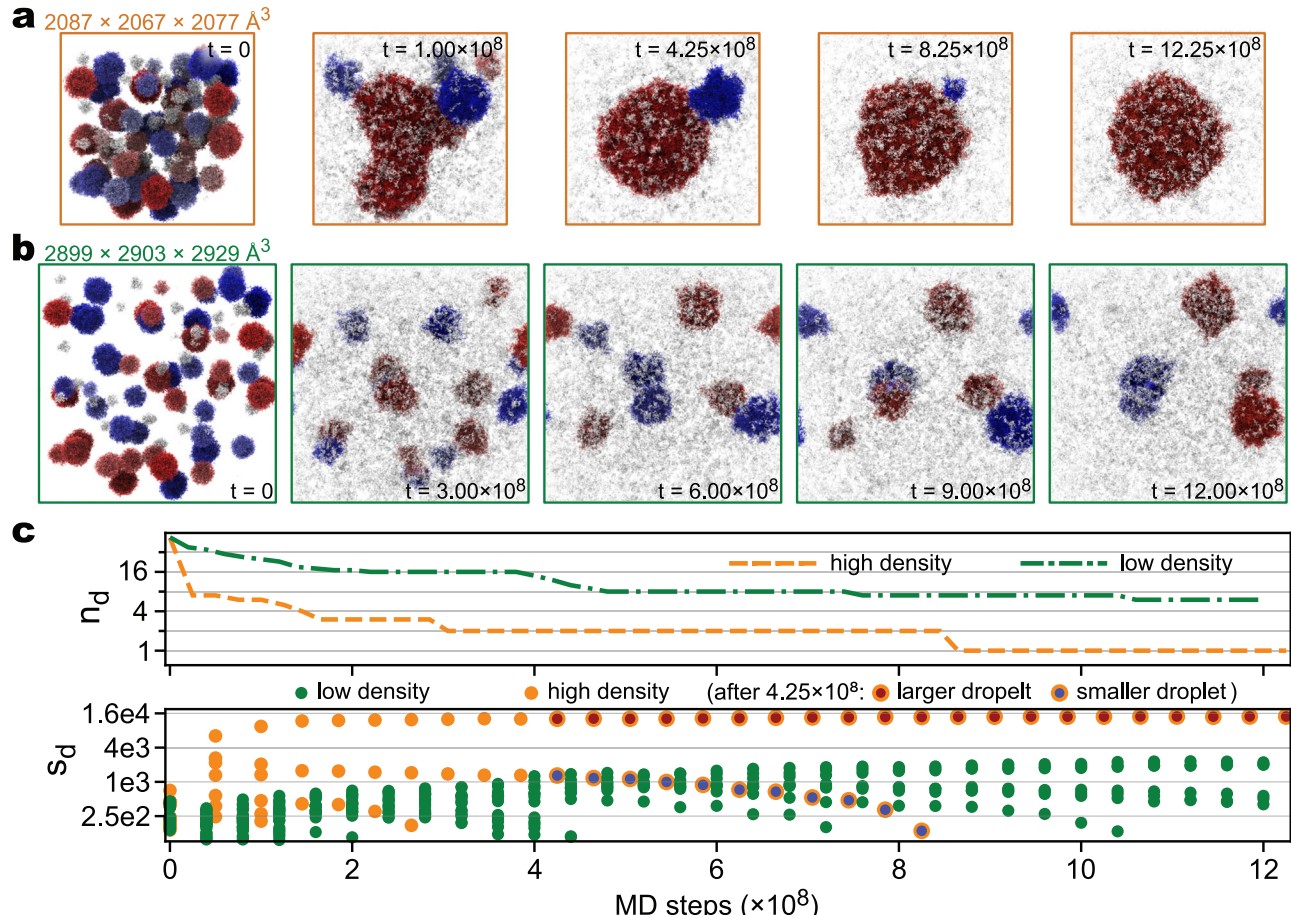

**Fig. 4 | CG MD simulations of the dynamics of multiple TDP-43-LCD droplets at two different densities. a** Snapshots of the high-density system taken at five different times during the simulation. **b** similar to (**a**) but for the low-density system. Colors of protein chains in (**a**) and (**b**) are according to the DBSCAN clustering results: chains in droplets are colored blue or red, and chains classified as "noise" are colored white. **c** Number of droplets ($n_d$, upper) and sizes of droplets ($s_d$, lower) as functions of simulation time. In the lower plot, special markers are used for the two droplets in the high-density system (**a**) after $4.25 \times 10^8$ steps: the markers have yellow edges, with the face color being red for the larger droplet and blue for the smaller droplet, respectively. For the other dots and lines, yellow represents the high-density system, while green represents the low-density system. Source data are provided as a Source Data file.

demonstrated in the current study, CGDYN is a powerful tool for simulating realistic-sized biomolecular condensates. The versatility of CGDYN could facilitate the reproduction of many experimental results, such as fluorescence recovery after photobleaching[49] and optical tweezers[61], and sheds more light on the molecular mechanisms underlying the biomolecular condensates.

We have developed CGDYN to enhance and broaden the applicability of CG MD simulations. ATDYN, an engine utilized in our prior residue-level CG simulations, suffers from speed limitations, preventing us from achieving long time-scale observations. When we introduce a non-bonded potential function, it merely incorporates an additional neighbor list search module. Consequently, it lacks optimization for handling multiple non-bonded interactions with distinct potential terms. CGDYN is specifically designed to address these shortcomings present in ATDYN. CGDYN endeavors to integrate various non-bonded potential functions into a single neighbor list search module, except for interactions involving protein-DNA interactions (where Newton's action-reaction principle cannot be applied) and electrostatic interactions (which are handled separately by considering only charged particles). Consequently, CGDYN necessitates defining cutoff values in ascending order and generating neighbor lists within a unified module according to atom types (such as ordered/disordered protein regions, DNA phosphate/sugar/base, etc). We believe that the distinctive aspect of CGDYN, setting it apart

from existing software, lies in its utilization of this hierarchical arrangement of cutoff values in managing multiple non-bonded interactions.

Our primary focus in this development is currently limited to CPU-based computers. Nonetheless, we anticipate that GPUs could further enhance the speed of MD simulations by employing the same domain decomposition with dynamic load balancing. While there have been several GPU implementations of CG potentials, optimizing multiple functions on GPU cores will pose significant challenges. In cases where the system size is not substantial, allocating all the energy/force calculations and integrations to a single GPU may be feasible. However, when utilizing multiple nodes with GPUs, communication between different nodes becomes a more critical issue due to the reduced computation-to-communication ratio inherent in GPU usage.

The LLPS of biomolecules and the subsequent transitions from liquid to solid phases can be influenced by specific proteins[48] or RNA molecules[62,63]. In a recent study, we explored the regulation of TDP-43-LCD's condensation by Hero-11[50]. Through slab simulations simulated with ATDYN, we could investigate the effects of Hero-11 on the interactions and dynamics of TDP-43-LCD, including its potential influence on droplet fusion through charge distribution. We expect that by incorporating Hero-11 into the TDP-43-LCD droplet systems examined in this work, we can provide direct evidence for one of the

previously proposed mechanisms of surface effect and develop a deeper understanding of the regulation of biomolecular LLPS. Furthermore, as demonstrated in Fig. 4, surface tension plays a crucial role in regulating droplet morphology, which, unfortunately, cannot be fully captured by slab simulations due to the infinite curvature imposed by periodic boundary conditions. This underscores the necessity of using our high-performance implementation in CGDYN to simulate such systems.

The internal structures of droplets or mesoscopic assemblies composed of multi-component proteins/RNAs have attracted significant attention[64–68]. Computational and experimental studies have focused on investigating the multi-layered structures in multi-component systems[64–66]. Valency has been proposed as a key factor in determining the spatial positioning of each component[66]. The impact of layered distributions, such as reducing the surface tension and localizing higher-valency components at the core while allowing rapid exchange of lower-valency components in and out of condensate, has been discussed[66]. The dynamics of multi-component LLPS has been proposed to involve multiple phase transitions, including the first separation of condensates from the bulk solution and subsequent transitions within the high-density phase[69].

There are increasingly interesting mesoscopic biological phenomena to be simulated with large-scale CG MD simulations with residue-level CG models. Paraspeckles, which are condensates composed of RNAs and IDPs, are observed in the cellular nucleus of mammalian cells[70]. A triblock copolymer model has been proposed to explain the shell localization of RNA ends and the size of paraspeckles[71]. However, due to their large sizes, achieving residue-level descriptions of such biologically interesting systems has been challenging. Multi-dimensional information on chromatin is also accumulated, providing a deeper understanding of how gene expression is regulated via dynamic interactions involving nucleosomes, transcription factors, remodelers, RNA polymerases, and other factors[72–75]. Experimental studies have suggested the functional roles of liquid droplets formed by transcription factors, mediators, and RNA polymerases[73–75]. To understand these models structurally and epigenetically, the residue-level descriptions are necessary at the very least. The methods and software developed in this study could be important computational bases for understanding mesoscopic biological phenomena through long-time dynamics of the real-size simulation systems.

## Methods

### Potential functions of the residue-level CG models
In GENESIS CGDYN, we employ residue-level CG models with an approximate resolution of 10 heavy atoms per particle. At this level, each CG particle represents a single amino acid residue in proteins, while nucleic acids are represented by three particles per nucleotide, corresponding to the phosphate (P), sugar (S), and base (B) components.

### Protein models
Regarding proteins, we have incorporated two distinct CG models: one for folded domains (AICG2+[5]) and another for IDRs (HPS/KH[14]). These models can be incorporated for proteins comprising folded domains and IDRs.

The AICG2+ potential energy function is given by[5]:

$$V_{\text{AICG2+}}(\Gamma) = V_{\text{local}} + \sum_{(i,j) \in \text{native contacts}} E_{\text{G}\bar{\text{o}}}(r_{ij}) + \sum_{(i,j) \in \text{nonnative contacts}} E_{\text{exv}}(r_{ij}). \quad (3)$$

where $\Gamma$ is the conformation of protein, $V_{\text{local}}$ includes all bonded terms, $E_{\text{G}\bar{\text{o}}}(r_{ij})$ is the structure-based Gō potential, and $E_{\text{exv}}(r_{ij})$ is the potential from excluded volume interaction.

The local interaction term, $V_{\text{local}}$ in Eq. (3), is defined as[5]:

$$V_{\text{local}} = \sum_{b_i \in \text{bonds}} k_{b,i}(b_i - b_{i,0})^2 + \sum_{r_i \in 1-3 \text{pairs}} -\varepsilon_i \exp\left(\frac{-(r_i - r_{i,0})^2}{2w_i^2}\right)$$
$$+ \sum_{\theta_i \in \text{angles}} -k_B T \ln \frac{P_\theta(\theta_i|i)}{\sin \theta_i}$$
$$+ \sum_{\varphi_i \in \text{dihedrals}} -\epsilon_{\varphi,i} \exp\left(\frac{-(\varphi_i - \varphi_{i,0})^2}{2\sigma_{\varphi,i}^2}\right)$$
$$+ \sum_{\varphi_i \in \text{dihedrals}} -k_B T \ln P_d(\varphi_i|i), \quad (4)$$

where the first term is for the bond interaction, the second term is for every two end particles in the angle one, and the fourth term is for the dihedral angle potential. The third and fifth terms are statistical flexible potentials, where $P_\theta(\theta_i|i)$ and $P_d(\varphi_i|i)$ are residue-type dependent probability distributions of angles and dihedral angles, respectively. $k_B$ is the Boltzmann constant and $T$ is temperature.

$E_{\text{G}\bar{\text{o}}}(r_{ij})$ in Eq. (3) is given by:

$$E_{\text{G}\bar{\text{o}}}(r_{ij}) = \varepsilon_{\text{G}\bar{\text{o}},i,j}\left[5\left(\frac{\sigma_{ij}}{r_{ij}}\right)^{12} - 6\left(\frac{\sigma_{ij}}{r_{ij}}\right)^{10}\right], \quad (5)$$

where $r_{ij}$ is the distance between residues forming a native contact, $\sigma_{ij}$ is the reference value of $r_{ij}$, and $\varepsilon_{\text{G}\bar{\text{o}},i,j}$ is the context-dependent energy coefficient. A native contact is defined as two residues having any heavy atoms within 4.5 Å from each other in the reference structure. Only the folded regions of proteins have $E_{\text{G}\bar{\text{o}}}(r_{ij})$ interactions.

$E_{\text{exv}}(r_{ij})$ in Eq. (3) is given by:

$$E_{\text{exv}}(r_{ij}) = \begin{cases} -\varepsilon_{\text{exv}}\left(\frac{\sigma_{ij}}{r_{ij}}\right)^{12} + \epsilon'_{\text{exv},ij}, & r_{ij} < r_C \\ 0, & r_{ij} \geq r_C \end{cases} \quad (6)$$

where $r_{ij}$ is the distance between residues $i$ and $j$, $\sigma_{ij}$ is residue-type dependent excluded volume distance[7], $\varepsilon_{\text{exv}} = 0.6 \text{kcal/mol}$ is the force coefficient, $r_C = 2\sigma_{ij}$ is the cutoff distance, and $\epsilon'_{\text{exv},i,j} = \left(\frac{1}{2}\right)^{12}\varepsilon_{\text{exv}}$.

The potential energy function of the HPS model is defined as[14]:

$$V_{\text{HPS}}(\Lambda) = \sum_{b_i \in \text{bonds}} E_b(b_i) + \sum_{(i,j) \in \text{non-bonded pairs}} E_{\text{HPS}}(r_{ij})$$
$$+ \sum_{(i,j) \in \text{charged pairs}} E_{\text{ele}}(r_{ij}), \quad (7)$$

where $\Lambda$ is the conformation of an IDP and $E_b(b_i)$ is the potential for every two neighboring particles with a bond length $b_i$:

$$E_b(b_i) = k_b(b_i - b_{i,0})^2, \quad (8)$$

where $b_{i,0} = 3.8 \text{Å}$ is the reference value and $k_b = 2.39 \text{kcal/mol} \cdot \text{Å}^{-2}$ is the force constant.

$E_{\text{HPS}}(r_{ij})$ is the interaction between non-bonded particles[14]:

$$E_{\text{HPS}}(r_{ij}) = \begin{cases} E_{\text{LJ}}(r_{ij}) + (1 - \lambda_{ij})\epsilon, & r_{ij} \leq 2^{1/6}\sigma_{ij} \\ \lambda_{ij}E_{\text{LJ}}(r_{ij}), & r_{ij} > 2^{1/6}\sigma_{ij} \end{cases} \quad (9)$$

where $\lambda_{ij}$ is the hydropathy and $E_{LJ}(r_{ij})$ is the Lennard−Jones potential:

$$E_{\text{LJ}}(r_{ij}) = 4\epsilon\left[\left(\frac{\sigma_{ij}}{r_{ij}}\right)^{12} - \left(\frac{\sigma_{ij}}{r_{ij}}\right)^6\right]. \quad (10)$$

In Eqs. (9) and (10), $\epsilon = 0.2$ kcal/mol. We use an arithmetic combinational rule for $\sigma_{ij}$ and $\lambda_{ij}$.

We utilize the Debye-Hückel term for the electrostatic interaction $E_{ele}(r_{ij})$:

$$E_{ele}(r_{ij}) = \frac{q_i q_j e^{-r_{ij}/\lambda_D}}{4\pi\varepsilon_0 \varepsilon_r r_{ij}}, \tag{11}$$

where $r_{ij}$ is the distance between two non-bonded charged particles $i$ and $j$ and $\varepsilon_0$ is the dielectric permittivity of vacuum. $\varepsilon_r$, the relative permittivity of the solution, is defined as a function of the temperature $T$ and salt molarity $C$: $\varepsilon_r = e(T)a(C)$, where $e(T) = 249.4 - 0.788T + 7.20 \times 10^{-4} T^2$ [76] and $a(C) = 1 - 0.2551C + 5.151 \times 10^{-2}C^2 - 6.889 \times 10^{-3}C^3$ [77]. The Debye length $\lambda_D$ is given by $\lambda_D = \sqrt{\frac{k_B T \varepsilon_0 \varepsilon_r}{2N_A e_c^2 I}}$, where $e_c$ is the elementary charge, $N_A$ is the Avogadro's number, and $I$ is the ionic strength of the solution.

## Nucleic acid models

We utilize the 3SPN.2 C model for DNA, whose potential energy function is defined as[18,19]:

$$\begin{aligned}
V_{3SPN.2C}(\Gamma) = &\sum_{b_i \in bonds} E_{bond}^{(2)}(b_i) + \sum_{\theta_i \in angles} E_{angle}^{(1)}(\theta_i) \\
&+ \sum_{\phi_i \in alldihedrals} E_{dihedral}^{(1)}(\phi_i) + \sum_{\phi_i \in backbone\ dihedrals} E_{dihedral}^{(2)}(\phi_i) \\
&+ \sum_{r_i \in exv\ pairs} E_{exv}^{(1)}(r_i) + \sum_{r_i \in phosphate\ pairs} E_{ele}(r_i) + \sum_{base\ steps} E_{bstk} \\
&+ \sum_{base\ pairs} E_{bp} + \sum_{cross-stacking\ pairs} E_{cstk},
\end{aligned} \tag{12}$$

where $\Gamma$ represents the conformation of DNA. Here, "excluded volume pairs" (shown as exv pairs in Eq. (12)) are the nonbonded particle pairs that do not participate in base pairing or stacking interactions.

Bond potential in Eq. (12) is defined as[18,19]:

$$E_{bond}^{(2)}(b_i) = k_{b,i}^{(2)}(b_i - b_{i,0})^2 + k_{b,i}^{(3)}(b_i - b_{i,0})^4, \tag{13}$$

where $b_i$ is the bond length, $b_{i,0}$ is the reference value of $b_i$, $k_{b,i}^{(2)}$ and $k_{b,i}^{(3)}$ are the force constants in the quadratic and quartic terms, respectively.

The angle potential in Eq. (12) is given by[18,19]:

$$E_{angle}^{(1)}(\theta_i) = k_{a,i}(\theta_i - \theta_{i,0})^2, \tag{14}$$

where $\theta_i$ is the bond angle formed by three CG particles, $\theta_{i,0}$ is the reference value, and $k_{a,i}$ is the force constant.

The dihedral angle potentials in Eq. (12) are defined as[18,19]:

$$E_{dihedral}^{(1)}(\phi_i) = \sum_n k_{\phi,i,n}[1 + \cos(n(\phi_i - \phi_{i,0}))], \tag{15}$$

and:

$$E_{dihedral}^{(2)}(\phi_i) = -\epsilon_{\phi,i} \exp\left(\frac{-(\phi_i - \phi_{i,0})^2}{2\sigma_{\phi,i}^2}\right), \tag{16}$$

where $\phi_i$ and $\phi_{i,0}$ are the dihedral angle and its reference value, respectively, $n$ is an integer number that controls the periodicity of the function, $\sigma_{\phi,i}$ is the Gaussian width, and $k_{\phi,i,n}$ and $\epsilon_{\phi,i}$ are the force constants.

The terms $E_{bstk}$, $E_{bp}$, and $E_{cstk}$ in Eq. (12) refer to multi-body energy functions describing base-base interactions[18,19]:

$$E_{bstk} = E_{Morse}^{(rep)}(r_i) + f(\Delta\theta_{BS,i})E_{Morse}^{(attr)}(r_i), \tag{17}$$

$$E_{bp} = E_{Morse}^{(rep)}(r_i) + \frac{1}{2}(1 + \cos\Delta\phi_{BP,i})f(\Delta\theta_{1,i})f(\Delta\theta_{2,i})E_{Morse}^{(attr)}(r_i), \tag{18}$$

$$E_{cstk} = f(\Delta\theta_{3,i})f(\Delta\theta_{CS,i})E_{Morse}^{(attr)}(r_i), \tag{19}$$

where $r_i$ represents the distance between the two interacting bases, and the angles ($\theta_{BS,i}$, $\theta_{1,i}$, $\theta_{2,i}$, $\theta_{3,i}$, and $\theta_{CS,i}$) and dihedral angles ($\phi_{BP,i}$) are formed by the surrounding sugar and phosphate sites. The Morse potential in Eqs. (17), (18), and (19) are defined as:

$$E_{Morse}(r_i) = \epsilon_{M,i}\left(1 - e^{-\alpha_i(r_i - r_{i,0})}\right)^2 - \epsilon_{M,i}, \tag{20}$$

where $r_i$ and $r_{i,0}$ are the distance between two particles and its reference value, respectively. $\epsilon_{M,i}$ and $\alpha_i$ are the "depth" and the "width" of the Morse potentials, respectively. The repulsive ($E_{Morse}^{(rep)}$) and the attractive ($E_{Morse}^{(attr)}$) components of the Morse potential are defined in the following:

$$E_{Morse}^{(rep)}(r_i) = \begin{cases} \epsilon_{M,i}\left(1 - e^{-\alpha_i(r_i - r_{i,0})}\right)^2, & r_i < r_{i,0} \\ 0, & r_i \geq r_{i,0} \end{cases} \tag{20a}$$

and

$$E_{Morse}^{(attr)}(r_i) = \begin{cases} -\epsilon_{M,i}, & r_i < r_{i,0} \\ \epsilon_{M,i}\left(1 - e^{-\alpha_i(r_i - r_{i,0})}\right)^2 - \epsilon_{M,i}, & r_i \geq r_{i,0} \end{cases} \tag{20b}$$

The angle-dependent modulating function in Eqs. (17), (18), and (19) is defined as[18,19]:

$$f(\Delta\theta) = \begin{cases} 1, & |\Delta\theta| < \gamma \\ 1 - \cos^2\left(\frac{\pi}{2\gamma}\Delta\theta\right), & \gamma \leq |\Delta\theta| \leq 2\gamma \\ 0, & |\Delta\theta| \geq 2\gamma \end{cases} \tag{21}$$

where $\Delta\theta$ is the difference between an angle ($\theta$) and its reference value, and $\gamma$ controls the tuning range.

The excluded volume interaction in Eq. (12) is given by[18,19]:

$$E_{exv}^{(1)}(r_i) = \begin{cases} \epsilon_{exv,i}\left[\left(\frac{\sigma_i}{r_i}\right)^{12} - 2\left(\frac{\sigma_i}{r_i}\right)^6\right] + \epsilon_{exv,i}, & r < r_{C,exv} \\ 0, & r \geq r_{C,exv} \end{cases} \tag{22}$$

where $r_{C,exv}$ is the cutoff distance and has the same value as $\sigma_i$.

For the electrostatic interactions in Eq. (12), we use the same definition given by the Debye-Hückel model as in Eq. (11).

For RNA, we offer two models: a structure-based model (similar to Eq. (3)) with a three-bead-per-nucleotide resolution[78], and an HPS model (Eq. (7)) with a one-bead-per-nucleotide resolution[37].

## Protein-DNA models

Protein-DNA binding can be divided into two types: sequence-nonspecific interactions involving amino acids and DNA backbone groups (primarily electrostatic interactions), and sequence-specific interactions between amino acids and DNA bases. The PWMcos model

can describe the latter by incorporating position weight matrix (PWM) information into structure-based interactions[79]. This model identifies a set of DNA-binding protein residues (DB-$C_\alpha$s) that contact DNA in its native structure. The potential energy is then calculated using:

$$\sum_{i\in bases}\sum_{j\in DB-C_\alpha}\sum_{m\in PWM\ columns} E_{PWMcos}(i,j,m,\vec{x})$$
$$= \sum_{i,j,m}\left(U_{m,j}(b_i,\vec{x}) + U_{m',j}(b_i,\vec{x})\right) \tag{23}$$

where $m'$ is the base in the complementary base of $m$, $b_i$ is the base type of base $i$ ($b_i \in [A,C,G,T]$), and $\vec{x}$ is the coordinates of particles in each conformation.

Function $U_{m,j}(b_i,\vec{x})$ is defined as[79]:

$$U_{m,j}(b_i,\vec{x}) = E_{Gaussian}(r_{ij})f(\Delta\theta_1)f(\Delta\theta_2)f(\Delta\theta_3), \tag{24}$$

where $r_{ij}$ is the distance between the $i$-th base and the $j$-th $C_\alpha$, and $\theta_1$, $\theta_2$, and $\theta_3$ are angles defined by the surrounding particles[79].

The Gaussian potential is defined as[79]:

$$E_{Gaussian}(r_i) = -\epsilon_{G,i}\exp\left(\frac{-(r_i - r_{i,0})^2}{2w_i^2}\right), \tag{25}$$

where $\epsilon_{G,i}$ and $w_i$ are the "depth" and "width" of the Gaussian, respectively. $\epsilon_{G,i}$ is a constant dependent on base type and the PWM. The modulating functions $f(\Delta\theta)$ is defined in Eq. (21).

Similar to PWMcos, we also implement a sequence-nonspecific model to describe the hydrogen bond (HB) interactions formed between protein and DNA backbone[10].

## Modeling of TDP-43-LCD

We used the AlphaFold2[80] predicted structure of TDP-43 as a reference structure for the residue-level CG modeling. In this modeling approach, each amino acid was represented by a single CG particle. We employed the HPS model[14] for simulating TDP-43-LCD (residues 261-319 and 335 to 414), mostly as an IDP. To preserve the secondary structure of an α-helix (residues 320 to 334), we employed the AICG2+ model[5]. The GENESIS-CG-tool is used to prepare all the structure and topology files for the MD simulations[35].

## Preparation of initial structures for the droplet systems

We constructed the initial structures for the droplet simulations in a stepwise manner. First, we simulated a single chain of TDP-43-LCD for $2\times10^7$ steps. From this simulation trajectory, we randomly selected a structure, which was then duplicated to create a system consisting of 500 chains. Subsequently, we performed "shrinking" simulations, gradually compressing the simulation boxes to dimensions of 500Å × 500Å × 500Å. The system was equilibrated at 260 K for $7\times10^6$ steps, resulting in the formation of a single droplet of TDP-43-LCD. Next, we selected structures obtained from the single-droplet simulations and put them into a larger simulation box to construct multiple-droplet systems.

**Specifically, we constructed the following systems.** We constructed a system consisting of 500 TDP-43-LCD chains and 100 Hero11 chains (Fig. 1c). We selected one simulated structure of the single TDP-43-LCD droplet system and placed it randomly within a simulation box of 1000Å × 1000Å × 1000Å. 100 Hero11 chains were then randomly added to the empty space, ensuring no structure clashes.

To construct the multiple-droplet systems in Figs. 2 and 4, we first carried out MD simulations of meso-scale particles with radii ranging from 50 Å to 200 Å. Only excluded volume interactions were considered during these simulations. Various numbers of large particles were simulated within simulation boxes of different sizes to achieve varying densities. Next, we superimposed the single-droplet TDP-43-LCD structures onto the large particles and removed any TDP-43-LCD chains that were located beyond the boundaries of the large particles. This process allows us to obtain multiple-droplet systems without structure conflicts.

To construct the initial structures of the two droplet simulations (Fig. 3), we chose two structures (named $\Lambda_1$ and $\Lambda_2$) from the single-droplet simulations of TDP-43-LCD. These two selected structures were placed into a simulation box with dimensions of 1000Å × 1000Å × 1500Å, ensuring that the two droplets in each structure were positioned close to each other with no direct contact (Fig. 3a). In the merged structure, certain chains in the dilute phase of $\Lambda_1$ overlapped with the condensate in $\Lambda_2$. For these chains, we relocated them to random locations in the dilute phase.

## Preparation of DNA structures for the benchmark systems

We first utilized the DNA structure building tool in GENESIS-CG-tool to generate a 200-bp double-stranded DNA (dsDNA) structure with a random sequence. Using the 3SPN.2 C model[18,19], the 200-bp dsDNA system consists of 1198 particles. Subsequently, we duplicated this dsDNA structure $n^2$ ($n=1,2,3,4,5$) times using GENESIS-CG-tool.

## Validation and benchmark simulations

For the benchmark tests, we first prepared droplet systems of TDP-43-LCD chains with four different particle numbers: (1) $N_1 = 300,146$, (2) $N_2 = 744,128$, (3) $N_3 = 1,190,882$, and (4) $N_4 = 2,565,178$. In all systems, there are two densities: $\rho_L = 6.74\times10^{-7}$ chains/Å$^3$ and $\rho_H = 1.85\times10^{-6}$ chains/Å$^3$.

Comparison tests between ATDYN (the atomistic decomposition method without dynamic load balancing), SPDYN-like (the midpoint cell method without dynamic load balancing), and CGDYN (the cell-based kd-tree domain decomposition with dynamic load balancing) are carried out for the systems with $N_1$ particles on Fugaku with 4 MPIs per node. Performances are investigated by checking the wall time of 10,000 MD steps. We save the trajectory files at the final step to mimic the real MD simulations. In all cases, we run the same runs five times and get the average as the benchmark results. The effect of the load balancing update during MD simulations is checked for $N_1$ particles with $\rho_H$ density. In this case, we ran $10^8$ MD steps and saved trajectories every 50,000 steps. In all cases, we assigned 35 Å as the electrostatic cutoff values ($r_{c,ele} = 35$Å)

Benchmark comparisons among ATDYN, CGDYN, and Open3SPN2 are carried out with the above-mentioned duplicated dsDNA systems. These systems have the cutoff distance $r_{c,ele} = 50$ Å. For ATDYN and CGDYN, we used Intel Xeon Gold 6242 CPUs (32 cores per node). For the test with Open3SPN2, Nvidia RTX A6000 GPU cards are used. Note that we were unable to perform simulations of our $n^2(n\geq2)$ dsDNA systems using the default Open3SPN2 software[51]. We had to modify the structure-reading component of Open3SPN2 to make it functional. However, we did not make any changes to the kernel part responsible for force/energy calculations, so the performance should remain consistent with what was reported in the original paper[51]. In addition, we faced memory issues when utilizing Open3SPN2 for the duplicated dsDNA systems with $n^2(n\geq6)$ 200-bp dsDNAs. However, we encountered no problems with such systems when using GENESIS ATDYN and CGDYN.

For the comparison between CGDYN and ATDYN for three systems on the RIKEN Hokusai supercomputer, we used the same working conditions as before[35].

To compare CGDYN with GROMACS, we generated two 1000 DPPC micelle systems with different densities. One micelle is generated by CHARMM-GUI Martini Maker[81], which has 100 DPPC molecules. We then duplicated this molecule and put in different places with different orientations. The finally generated systems have 1.2 million particles with two system sizes: 2377.059 Å × 2395.0 Å × 2395.0 Å and

1598.128 Å × 1598.767 Å × 1597.518 Å. We used 11 Å as the cutoff value. We estimated the performance by running 5000 MD steps with 0.03 ps time step. We used 2023.1 version of GROMACS for benchmark comparison.

In the benchmark comparison of ATDYN, CGDYN, and LAMMPS, we simulated a multiple IDP droplets system consisting of 300,146 particles (Fig. 2a, right). For LAMMPS, we utilized the version dated 29-Oct-2020 and incorporated the implementation of the HPS model (https://github.com/azamat-rizuan/HPS-SS-model). For both GENESIS (ATDYN and CGDYN) and LAMMPS, cutoffs of 20 Å for HPS and 35 Å for Debye-Hückel potentials were applied. Additionally, a buffer size of 3 Å was employed for pairlist generation. All benchmark simulations were conducted using Langevin dynamics at 300 K, with a friction coefficient of 0.01 ps$^{-1}$. The solution temperature was set to 300 K and the ionic strength to 150 mM in GENESIS, with the dielectric constant accordingly set to 74.911 in LAMMPS. For load balancing in LAMMPS, the RCB algorithm was used every 1000 steps, with a load imbalance threshold set at 1.1. In CGDYN, load balancing was performed during the initialization of calculations.

### CG MD simulations of droplet dynamics

All CG simulations were performed using CGDYN. In all the simulations, we used a time-integration step size of 10 fs. Nonlocal HPS ($E_{HPS}$) and electrostatic ($E_{ele}$) potentials have cutoffs of 20 Å and 35 Å, respectively. Production runs were conducted in NVT ensembles, using Langevin dynamics with a friction coefficient of $0.01ps^{-1}$.

The two-droplet simulations were conducted in the periodic boundary condition (PBC) with box dimensions of 1000Å × 1000Å × 1500Å. MD simulations were carried out at 280 K for $5 \times 10^8$ steps. However, we found that the fusion process occurred within a relatively short time during the simulations. Therefore, only the first $1 \times 10^8$ steps data are analyzed and shown in the current study. The results are consistent across all five independent runs (Fig. 3g, h, and Supplementary Fig. 9). We performed the simulations on local PC clusters (Intel Xeon Gold 6240 R CPU, 2.4 GHz) using 16 MPI processes in conjunction with 3 OpenMP threads.

The ultra-large multiple-droplet simulations were conducted in PBC boxes of 2087Å × 2067Å × 2077Å (high-density) and 2899Å × 2903Å × 2929Å (low-density). We performed these simulations at $T = 290K$. The high-density and low-density systems were simulated for $12.25 \times 10^8$ and $12.00 \times 10^8$ steps, respectively. These simulations were conducted on the supercomputer Fugaku using 512 or 1024 nodes.

### Data analysis

The DBSCAN clustering method[54] was employed to analyze the structures from the droplet simulations. However, different parameters were used for the two-droplet and multiple-droplet systems. In the two-droplet system, we defined the contact distance between two chains as $d_{DBSCAN,2-drop} = 1$ when the two chains formed contacts (i.e., when two inter-chain residues were within 10 Å from each other), and 0 otherwise. On the other hand, in the multiple-droplet system, we define the contact distance between two chains as the distance between their center of mass (COM). With these definitions, we used $\varepsilon = 0.5$ (radius of neighborhood), min_pts = 20 (the minimum number of neighbors for a point to be considered as a core point), and min_cluster_size = 100 (minimum cluster size) in the two-droplet systems. For the multiple-droplet systems, we used $\varepsilon = 50$Å, min_pts = 5, and min_cluster_size = 50. These parameters were chosen to effectively identify clusters and analyze the droplet structures in each system.

To monitor the mixing of contents in the two-droplet system, we define a coordinate $D_{IJ}$ to describe the average distance

between every two chains, one coming from cluster $I$ and the other from cluster $J$:

$$D_{IJ} = \frac{\sum_{i \in C_I, j \in C_J, i \neq j} d_{COM,ij}}{N_{IJ}}, \tag{26}$$

where $I$ and $J$ can be 1 or 2 (one of the two clusters in the initial structure), $N_{IJ} = \sum_{i \in C_I, j \in C_J, i \neq j} 1$ is the total number of computed chain pairs. On top of $N_{IJ}$, we defined a "mixing" coordinate $m$ as:

$$m = \frac{\sqrt{D_{1,1}D_{2,2}}}{D_{1,2}}. \tag{27}$$

We employed a coordinate $\eta$ to describe the shape of a droplet:

$$\eta = \max\left(\frac{d_x}{d_y}, \frac{d_y}{d_x}\right) + \max\left(\frac{d_y}{d_z}, \frac{d_z}{d_y}\right) + \max\left(\frac{d_z}{d_x}, \frac{d_x}{d_z}\right). \tag{28}$$

### Reporting summary

Further information on research design is available in the Nature Portfolio Reporting Summary linked to this article.

## Data availability

Benchmark and simulation data generated in this study have been deposited at https://github.com/RikenSugitaLab/cgdyntest/, where analysis scripts are also deposited. Initial, final, and some intermediate structures from our simulations are stored on the same GitHub repository. Given that the total volume of MD trajectories amounts to 7.0 TB, we have not uploaded them to public repositories. However, we ensure full reproducibility of our results with the provided data (MD control files, initial structures, and analysis scripts). Additionally, MD trajectories are available upon request to the corresponding author. Source data are provided as a Source Data file. Source data are provided with this paper.

## Code availability

The GENESIS CGDYN code and analysis programs can be found at https://github.com/genesis-release-r-ccs/genesis-2.1.0beta_cgdyn[82]. The GENESIS-CG-tool for creating CGDYN inputs is available at https://github.com/genesis-release-r-ccs/genesis_cg_tool. Additionally, in-house scripts utilized for analyzing droplet systems are hosted at https://github.com/RikenSugitaLab/cgdyntest.

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

## Acknowledgements

This work was supported in part by MEXT JSPS Kakenhi (grant number 19H05645, 21H05249 (to Y.S.), 21H05282 (to J.J. and C.T.)), RIKEN pioneering projects "Biology of Intracellular Environments", and "Glycolipidologue Initiative" (to Y.S.), RIKEN incentive (to J.J. and C.T.), MEXT program for promoting research on the supercomputer Fugaku (JPMXP1020200101), and MEXT program for Big-data-driven bio/synthetic polymer science to create absolutely circular materials (JPMXP1122714694) and Data-Driven Research Methods Development and Materials Innovation Led by Computational Materials Science (JPMXP1020230327) (to Y.S.). The computer resources are provided by the HPCI system research project (Project ID: ra000003, hp200028, hp200135, hp210177, hp220170, hp230072, and hp230212) and by RIKEN Advanced Center for Computing and Communication (for HOKUSAI BigWaterfall, project Q22535, Q22536, and Q23536).

## Author contributions

J.J. developed the program, prepared a part of the benchmark system, performed benchmark tests, and ran MD simulations. C.T. prepared benchmark and simulation systems, ran MD simulations, and analyzed simulation trajectories. Y.S. planned and organized the overall research. All authors contributed the preparation of the manuscript.

## Competing interests

The authors declare no competing interest.
