## [Peer Review File · Nature Communications]

GENESIS CGDYN: large-scale coarse-grained MD simulation with dynamic load balancing for heterogeneous biomolecular systemsReviewers' Comments:

Reviewer #1:

Remarks to the Author:

In this paper, Jung et al. present and benchmark a new domain decomposition scheme (DDS) to accelerate and facilitate molecular dynamics (MD) simulations using coarse-grained (CG) implicit solvent models. After providing a brief description of the new DDS, they illustrate its power by simulating three variations of protein droplet fusion simulations of increasing complexity with the largest test system nearly reaching experimental length scales. Additionally, the Supporting Information offers a performance comparison to Open3SPN package for DNA simulations.

In general, inefficient DDSs for CG force fields are an issue when running simulations with engines that are primarily optimized for the simulation of all-atom force fields. Whereas the influence is usually not so large for CG models that utilize explicit water models, it becomes a significant problem for implicit water models. Likewise, the DDS can become a problem when systems with different regions of density are simulated like for example a liquid-vapor interface.

However, the idea of dynamic load-balancing and heterogeneous domain decomposition is not new and is implemented for a number of packages. Some of these packages are used for both CG simulations and atomistic resolutions.[1], [2] Other packages even have dedicated CG implementations.[3], [4] Given that the core of this paper focuses on presenting such a new DDS, I'm surprised by the lack of any mention of other similar schemes both in the introduction and discussion. For example, ddcMD[3], [5] was specifically developed with heterogeneous domain decomposition in mind. Other programs such as GROMACS[2] or ESPResSo[4] provide rather efficient DDSs and dynamic load-balancing as well. Even LAMMPS[1] supports some ways to optimize the domain decomposition even though it appears not to support dynamic rescaling during simulation. I recommend that the authors give a brief but complete summary of schemes used in other software packages.

Either in the introduction or discussion, they should also mention why the new scheme is 'novel'. One could even argue that the GROMACS scheme is more advanced since it monitors force load imbalance during the simulation and adjusts DLB accordingly on the fly. In fact, I fail to see why the presented scheme is better than any of the other schemes. For this reason, I strongly recommend removing the phrase 'novel' (see line 106) from the manuscript. The term 'unique' which is used in line 86 seems more appropriate.

Furthermore, in the introduction, the authors mention the "incorporation of diverse potential functions" adding to the complexity of optimizing CG MD software. This problem is a very valid one but not considered in the remainder of the paper. This issue could be further discussed in the discussion. It would be interesting to know how CGDYN tackles this particular challenge.

With the previously mentioned statement, the authors reference an earlier paper[6] of their group. That paper deals with the implementation of the here-used CG models and their potential in the GENESIS program. In my view, it is a bit confusing as to what aspects of CG simulations have already been published as part of that paper and which new features CGDYN brings to the table. For this reason and because this paper concerns a new algorithm, I believe some text on how CGDYN is integrated into the GENESIS package would be appropriate. It should also mention if there are any other new features aside from the DDS.

Overall, I have not much to critique in the results section. However, one could argue that the section 'CGDYN structure' from the methods section should be moved to the results since it really is part of the implementation that is the focus of the paper. If done so it would set the stage for discussing how the cut-offs of different forces and the DLB play together.

Furthermore, comparing the DLB scheme to other MD engines would certainly strengthen the claim of novelty made in this paper. At the moment the authors provide a comparison to Open3SPN only in the SI. However, I see some problems with this comparison. On the one hand, GENESIS runs on CPUs only while OpenMM is GPU-accelerated. Secondly, to my knowledge, OpenMM does not employ a dynamic load balancing. Thus, my suggestion would be to use a simpler CG model that runs on more than just a single engine. For example, using the dry Martini force field[7] (i.e. with implicit water) would be a more suitable target. Due to its simplicity, the (dry) Martini force field runs on GROMACS[7], OpenMM[8], and ddcMD[3]. It utilizes an implicit solvent model, which means the particle size distribution is inhomogeneous. Additionally, when using the model to look at the self-assembly of large vesicles the particle size distribution can be expected to vary quite rapidly. Using such a test case and comparing 5 different engines (i.e. the previously mentioned three plus the two GENESIS ones) should yield a very compelling argument.

Overall, I'm not convinced that this paper is appropriate for the broad readership of Nature Communications. The presented algorithm targets coarse-grained simulations, which is a subfield of simulation techniques. Even within this subfield the DDS only appears to accelerate models using implicit solvent treatment. Furthermore, the presented CG models have already been previously implemented by the authors in the GENESIS package. Whereas the acceleration they present is solid when compared to this original implementation, it probably would have been possible to simulate the test systems with the old code. It would have taken longer but not necessarily prohibitively longer. Nevertheless, I'm convinced the implementation of this DDS in CGDYN will advance the field of CG modeling of large biomolecular systems. Provided, the above-mentioned concerns are addressed I see no reason why this paper should not be published in a more specialized journal. For example, Nature Computational Science seems to be a better fit for presenting a new computational method such as this one.

Minor Comments

The statement in line 188 a reference is missing.

The use of the term residue level CG models in the introduction is a bit misleading. It would seem more appropriate to split the mentioned CG models early on into those using implicit and explicit solvent models in order to make clear that the DDS mostly focuses on the first class of models.

References

- [1] A. P. Thompson et al., "LAMMPS - a flexible simulation tool for particle-based materials modeling at the atomic, meso, and continuum scales," *Comput Phys Commun*, vol. 271, p. 108171, Feb. 2022, doi: 10.1016/j.cpc.2021.108171.
- [2] S. Páll et al., "Heterogeneous parallelization and acceleration of molecular dynamics simulations in GROMACS," *J Chem Phys*, vol. 153, no. 13, Oct. 2020, doi: 10.1063/5.0018516.
- [3] X. Zhang et al., "ddcMD: A fully GPU-accelerated molecular dynamics program for the Martini force field," *J Chem Phys*, vol. 153, no. 4, Jul. 2020, doi: 10.1063/5.0014500.
- [4] H. V. Guzman, C. Junghans, K. Kremer, and T. Stuehn, "Scalable and fast heterogeneous molecular simulation with predictive parallelization schemes," *Phys Rev E*, vol. 96, no. 5, p. 053311, Nov. 2017, doi: 10.1103/PhysRevE.96.053311.
- [5] J.-L. Fattebert, D. F. Richards, and J. N. Glosli, "Dynamic load balancing algorithm for molecular dynamics based on Voronoi cells domain decompositions," *Comput Phys Commun*, vol. 183, no. 12, pp. 2608–2615, Dec. 2012, doi: 10.1016/j.cpc.2012.07.013.
- [6] C. Tan, J. Jung, C. Kobayashi, D. U. La Torre, S. Takada, and Y. Sugita, "Implementation of residue-level coarse-grained models in GENESIS for large-scale molecular dynamics simulations," *PLoS Comput Biol*, vol. 18, no. 4, p. e1009578, Apr. 2022, doi: 10.1371/journal.pcbi.1009578.
- [7] C. Arnarez et al., "Dry Martini, a Coarse-Grained Force Field for Lipid Membrane Simulations with Implicit Solvent," *J. Chem. Theory Comput.*, vol. 11, no. 1, pp. 260–275, 2015, [Online]. Available: <https://doi.org/10.1021/ct500477k>

[8] J. L. MacCallum, S. Hu, S. Lenz, P. C. T. Souza, V. Corradi, and D. P. Tieleman, "An implementation of the Martini coarse-grained force field in OpenMM," *Biophys J*, vol. 122, no. 14, pp. 2864–2870, Jul. 2023, doi: 10.1016/j.bpj.2023.04.007.

Reviewer #2:

Remarks to the Author:

Jung and coworkers introduce an extension of the Genesis molecular dynamics engine that aims to solve the load-balancing problem, an issue that is prevalent when simulating systems with heterogeneous particle distributions, e.g., implicit-solvent coarse-grained molecular dynamics. The new code is referred to as CGDYN and the authors demonstrate its utility for large simulations of droplets undergoing fusion. The computational experiments are well-designed and the analysis sound. The manuscript is also well-written.

The main novelty presented by the manuscript is the cell-based kd-tree method that is used to perform heterogeneous domain decomposition with dynamic rebalancing during the simulation. However, based on my understanding of the presented methodology, the cell-based kd-tree method should yield comparable results to existing dynamic load balancing methods. For example, GROMACS does iterative dynamic load balancing where each cell is independently resized and staggered relative to other cells. LAMMPS has the recursive coordinate bisectioning method that is similar to the CGDYN approach. While perhaps not as widespread of a codebase, ls1 mardyn implements a kd-tree based dynamic load balancing approach that is perhaps most similar to the authors' approach. In other words, the heterogeneity of implicit-solvent coarse-grained molecular dynamics simulations can be largely mitigated by existing dynamic load balancing methods. Therefore, it is unclear to me how CGDYN improves upon the state-of-the-art other than incrementally.

I am also surprised by the lack of comparison to existing load balancing methods. In the manuscript, the authors have only compared the performance of CGDYN to the group's other codes rather than any of the open-source MD engines mentioned in the introduction. While a quantitative comparison to other heterogeneous domain decomposition methods would be ideal, I understand that this may be a time-consuming process. At minimum, the authors should discuss CGDYN's approach in the context of the numerous load balancing methods that have been published that address the same issue of particle heterogeneity. I mentioned some above and have compiled an incomplete list below:

1. Seckler et al., "Load Balancing for Molecular Dynamics Simulations on Heterogeneous Architectures"
2. Grime et al., "Highly Scalable and Memory Efficient Ultra-Coarse-Grained Molecular Dynamics Simulations"
3. Fattebert et al., "Dynamic load balancing algorithm for molecular dynamics based on Voronoi cells domain decompositions"
4. Zhang et al., "Dynamic load balancing based on constrained k-d tree decomposition for parallel particle tracing"

Reviewer #3:

Remarks to the Author:

This paper presents a state-of-the-art implementation of simulation and modeling of coarse-grained macromolecular systems.

What is new:

It is specifically designed to be efficient for systems with very large conformational changes. It focuses

on efficiency for systems with unusual or weird particle distributions, as is often seen when implicit solvent is used.

What it is good for: Intrinsically disordered protein mixing. Very large mesoscopic simulations. One given example involves droplet formation, where homogeneous particles condense into a single dense droplet.

In the introduction, it should perhaps be noted that many CG simulation projects are also done using CHARMM. This is especially true for early CG simulations, and for those requiring Hessians. The list of seven programs seems incomplete with including CHARMM, for which there are many dozens of published CG projects.

A new domain decomposition scheme with dynamic load balancing.
The scalability of the new approach as shown by figure 2 is very good.

One outstanding questions relate to how well this approach will work on GPU based supercomputers. I believe the authors need to, at least, mention and address this issue and identify potential problems and difficulties that might arise for hardware that may become dominant in the future.

Recommendation: Publish with minor additions.

In this reply, our replies were highlighted in red, while the sentences in the revised manuscript were in blue.

Reviewer #1 (Remarks to the Author):

(1) In this paper, Jung et al. present and benchmark a new domain decomposition scheme (DDS) to accelerate and facilitate molecular dynamics (MD) simulations using coarse-grained (CG) implicit solvent models. After providing a brief description of the new DDS, they illustrate its power by simulating three variations of protein droplet fusion simulations of increasing complexity with the largest test system nearly reaching experimental length scales. Additionally, the Supporting Information offers a performance comparison to Open3SPN package for DNA simulations.

In general, inefficient DDSs for CG force fields are an issue when running simulations with engines that are primarily optimized for the simulation of all-atom force fields. Whereas the influence is usually not so large for CG models that utilize explicit water models, it becomes a significant problem for implicit water models. Likewise, the DDS can become a problem when systems with different regions of density are simulated like for example a liquid-vapor interface.

However, the idea of dynamic load-balancing and heterogeneous domain decomposition is not new and is implemented for a number of packages. Some of these packages are used for both CG simulations and atomistic resolutions.[1], [2] Other packages even have dedicated CG implementations.[3], [4] Given that the core of this paper focuses on presenting such a new DDS, I'm surprised by the lack of any mention of other similar schemes both in the introduction and discussion. For example, ddcMD[3], [5] was specifically developed with heterogeneous domain decomposition in mind. Other programs such as GROMACS[2] or ESPResSo[4] provide rather efficient DDSs and dynamic load-balancing as well. Even LAMMPS[1] supports some ways to optimize the domain decomposition even though it appears not to support dynamic rescaling during simulation. I recommend that the authors give a brief but complete summary of schemes used in other software packages.

We agree with the reviewer regarding the novelty of dynamic load balancing and heterogeneous domain decomposition written in our work. Our dynamic load balancing scheme is similar to the recursive coordinate bisection (RCB) algorithm in LAMMPS or kd-tree schemes employed in other MD programs. However, our scheme is applicable not only to simple molecular models and potential functions but also to more complex models, such as residue-level coarse-grained models for proteins, nucleic acids, and other molecular systems. In response to the reviewer's suggestion, we have briefly elaborated on the schemes utilized in other software packages in the introduction section of the revised manuscript. In the revised manuscript, we added the citations [31,41-47].

In the revised manuscript, on line 94, we added the following sentences:

Various endeavors have been undertaken to enhance the efficiency of MD simulations through the development of dynamic load-balancing schemes. LAMMPS employs the recursive coordinate bisection (RCB) algorithm for dynamic load balancing. In Gromacs, domain sizes are dynamically adjusted based on computational time for each process. The ddcmd program introduces domain decomposition based on Voronoi cells. The ESPResSo software utilizes domain decomposition based on space-filling curve, with dynamic re-balancing achieved through a collection of adaptive octrees. Guzman et al. proposed a domain decomposition scheme suitable for multi-scale simulations by assigning different domain sizes for all-atom and CG models. Additionally, Grime and Voth developed a highly scalable scheme for ultra-coarse-grained models using the Hilbert space-filling curve. Some have employed kd-tree schemes for dynamic load balancing.

(2) Either in the introduction or discussion, they should also mention why the new scheme is 'novel'. One could even argue that the GROMACS scheme is more advanced since it monitors force load imbalance during the simulation and adjusts DLB accordingly on the fly. In fact, I fail to see why the presented scheme is better than any of the other schemes. For this reason, I strongly recommend removing the phrase 'novel' (see line 106) from the manuscript. The term 'unique', which is used in line 86, seems more appropriate.

We removed the phrase "novel" in the revised manuscript and emphasized the uniqueness of our scheme.

In the revised manuscript, on line 125, we modified the following sentence:

We have developed a unique domain decomposition scheme with dynamic load balancing to parallelize the residue-level CG MD simulations.

(3) Furthermore, in the introduction, the authors mention the "incorporation of diverse potential functions" adding to the complexity of optimizing CG MD software. This problem is a very valid one but not considered in the remainder of the paper. This issue could be further discussed in the discussion. It would be interesting to know how CGDYN tackles this particular challenge.

The incorporation of diverse potential functions is one of the key points in our development of CGDYN. In the revised manuscript, we emphasized the unified pairlist scheme of different non-bonded interactions, such as excluded volume, DNA-base pairing, HPS, and so on. In ATDYN/GENESIS, each non-bonded interaction pairlist is generated in a different module since the cutoff length of each non-bonded interaction is significantly different. In contrast, CGDYN/GENESIS generates all non-bonded interaction pairlist in one module, except protein-DNA interaction using PWMCoS potential and electrostatic interaction. This unified pairlist scheme can save computational time to generate the pairlist in energy and force calculations. This is one of the key points

why CGDYN/GENESIS can outperform other MD programs in terms of computational performance.

In the revised manuscript, on line 110, we added the following sentence:

In addition, we implemented a united neighboring list search algorithm to calculate the energy and forces of diverse potential functions with different cutoff values in residue-level CG MD simulations, and thereby, CGDYN outperforms other MD programs in terms of computational performance.

In the revised manuscript, on line 428, we added the following sentences:

We have developed CGDYN to enhance and broaden the applicability of CG MD simulations. ATDYN, an engine utilized in our prior residue-level CG simulations, suffers from speed limitations, preventing us from achieving long time-scale observations. When we introduce a new non-bonded potential function, it merely incorporates an additional neighbor list search module. Consequently, it lacks optimization for handling multiple non-bonded interactions with distinct potential terms. CGDYN is specifically designed to address these shortcomings present in ATDYN. CGDYN endeavors to integrate various non-bonded potential functions into a single neighbor list search module, except for interactions involving protein-DNA interactions (where Newton's action-reaction principle cannot be applied) and electrostatic interactions (which are handled separately by considering only charged particles). Consequently, CGDYN necessitates defining cutoff values in ascending order and generating neighbor lists accordingly within a unified module according to atom types (such as ordered/disordered protein regions, DNA phosphate/sugar/base, etc). We believe that the distinctive aspect of CGDYN, setting it apart from existing software, lies in its utilization of this hierarchical arrangement of cutoff values in managing multiple non-bonded interactions.

(4) With the previously mentioned statement, the authors reference an earlier paper[6] of their group. That paper deals with the implementation of the here-used CG models and their potential in the GENESIS program. In my view, it is a bit confusing as to what aspects of CG simulations have already been published as part of that paper and which new features CGDYN brings to the table. For this reason and because this paper concerns a new algorithm, I believe some text on how CGDYN is integrated into the GENESIS package would be appropriate. It should also mention if there are any other new features aside from the DDS.

Our earlier paper [6] focuses on the implementations of residue-level coarse-grained models for proteins, intrinsically disordered regions (IDRs), DNA, RNA, and their complexes, whose interactions are computed with multiple potential energy functions. The GENESIS-CG-Tool can effectively generate input files of various CG models for biomolecules. Although a hybrid parallelization, a combination of MPI/OpenMP schemes, was developed for the residue-level CGMD simulations, the DDS was not discussed in ref. [6].

The lack of the DDS in our previous works limits the available size of CG MD simulation systems, which will be overcome in the current work. In the current paper, the DDS and a unified neighboring list search algorithm were implemented to calculate the energy and forces of diverse potential functions with different cutoff values much more efficiently. These innovations allow us to simulate biomolecular systems with larger sizes for longer times without reducing the complexity of biomolecular structures and interactions.

Please see the modified sentences written in (3).

(5) Overall, I have not much to critique in the results section. However, one could argue that the section 'CGDYN structure' from the methods section should be moved to the results since it really is part of the implementation that is the focus of the paper. If done so it would set the stage for discussing how the cut-offs of different forces and the DLB play together.

We moved the 'CGDYN structure' section from the Method to the Result section as suggested in the revised manuscript on pages 7-9.

(6) Furthermore, comparing the DLB scheme to other MD engines would certainly strengthen the claim of novelty made in this paper. At the moment the authors provide a comparison to Open3SPN only in the SI. However, I see some problems with this comparison. On the one hand, GENESIS runs on CPUs only while OpenMM is GPU-accelerated. Secondly, to my knowledge, OpenMM does not employ a dynamic load balancing. Thus, my suggestion would be to use a simpler CG model that runs on more than just a single engine. For example, using the dry Martini force field[7] (i.e. with implicit water) would be a more suitable target. Due to its simplicity, the (dry) Martini force field runs on GROMACS[7], OpenMM[8], and ddcMD[3]. It utilizes an implicit solvent model, which means the particle size distribution is inhomogeneous. Additionally, when using the model to look at the self-assembly of large vesicles the particle size distribution can be expected to vary quite rapidly. Using such a test case and comparing 5 different engines (i.e. the previously mentioned three plus the two GENESIS ones) should yield a very compelling argument.

As suggested by the reviewer, we added the performance comparisons with other software: the dry Martini using Gromacs and the HPS using LAMMPS. Since we have already shown the results of Open3SPN based on OpenMM, the performance comparisons between CGDYN/GENESIS and three other software were carried out. Since we had to implement new programs to compare the dry Martini simulations in CGDYN, this was our best effort during the short revision period. As shown in the revised manuscript, all the performance comparisons have shown the superiority of CGDYN/GENESIS. We believe that this comparison can indicate the usefulness of CGDYN for extensive biological simulations with the residue-level coarse-grained models.

In the revised manuscript, on line 271, we added the following sentences:

The comparison between CGDYN and Open3SPN2 regarding dynamic load balancing effects is not straightforward. Hence, we assessed the performance of GENESIS and Gromacs on Fugaku by creating clusters of DPPC micelles using the dry Martini force field. Supporting Fig. 8 reveals that Gromacs exhibits similar performance to CGDYN for 32 nodes. However, CGDYN surpasses Gromacs from 64 nodes onwards, and this performance margin widens with an increasing number of processes. Our primary aim in comparing the performance of the dry Martini system models is not to demonstrate the superiority of CGDYN within the dry Martini model but to ascertain whether its parallelization is comparable to existing MD software equipped with robust dynamic load balancing schemes. We also observed that the effect of dynamic load balancing within the dry Martini model is not pronounced as with the residue-level CG model. This is primarily attributed to the small cutoff distance in non-bonded interactions within the dry Martini model, resulting in a lower computation-to-communication ratio. We additionally evaluated performance for smaller systems (150,000 particles), wherein Gromacs exhibited superior performance to CGDYN from 16 to 128 nodes. Considering this, Gromacs appears to be more optimized than CGDYN for energy/force calculation, whereas CGDYN demonstrates superior scalability for larger systems with nonuniform particle densities.

(7) Overall, I'm not convinced that this paper is appropriate for the broad readership of NatureCommunications. The presented algorithm targets coarse-grained simulations, which is a subfield of simulation techniques. Even within this subfield the DDS only appears to accelerate models using implicit solvent treatment. Furthermore, the presented CG models have already been previously implemented by the authors in the GENESIS package. Whereas the acceleration they present is solid when compared to this original implementation, it probably would have been possible to simulate the test systems with the old code. It would have taken longer but not necessarily prohibitively longer. Nevertheless, I'm convinced the implementation of this DDS in CGDYN will advance the field of CG modeling of large biomolecular systems. Provided, the above-mentioned concerns are addressed I see no reason why this paper should not be published in a more specialized journal. For example, Nature Computational Science seems to be a better fit for presenting a new computational method such as this one.

The manuscript presents both technical innovations and biophysical findings. These include the DDS, DLB, and unified pairlist generation techniques in CGDYN, which collectively enhance CGDYN's performance beyond other MD software utilizing DDS. Furthermore, we highlight CGDYN's versatility in handling various biological applications through residue-level coarse-grained models on high-performance computing platforms. Such technical advancements are crucial for unlocking new molecular and cellular biology insights.

In the revised manuscript, we have tripled the duration of our protein droplet dynamics simulations from the previous length, reaching a new total of $\sim 1.2 \times 10^9$ steps (two systems). We observed both droplet fusion and dissolution events. In the high-density system, the system evolves into a single-droplet state. Interestingly, we noted that smaller droplets gradually shrank and eventually dissolved into the dilute phase, while larger droplets continued to grow (revised Figure 4). This process is reminiscent of Ostwald ripening observed in emulsions. This marks the first direct observation of this phenomenon in biomolecular systems using residue-level coarse-grained MD simulations. We believe this result will attract widespread attention in computational molecular biology.

Furthermore, many computational studies on biomolecular condensation have relied on slab simulations, which utilize a simulation box with two short dimensions and one long dimension, coupled with periodic boundary conditions, to expedite reaching equilibrium. However, the slab method fails to accurately represent the system size-dependent surface tension, which, as demonstrated in our simulations, plays a crucial role in determining the behavior of small droplets. This underscores the necessity of using our CGDYN for droplet simulations in biomolecular condensation studies.

We want to emphasize that such large-scale biological simulations, offering residue-level descriptions of biomolecules, were prohibitively expensive with our previous implementation (ATDYN). Thus, our current work paves a new pathway for bridging our understanding of molecular structures/interactions with sub-cellular dynamics.

In the revised manuscript, we have replotted Figure 4 using our new MD trajectories of 1.2×10^9 steps. Accordingly, the figure caption has been changed to:

Figure 4. CG MD simulations of the dynamics of multiple TDP-43-LCD droplets at two different densities. (a) Snapshots of the high-density system taken at five different times during the simulation. (b) similar to (a) but for the low-density system. Colors of protein chains in (a) and (b) are according to the DBSCAN clustering results: chains in droplets are colored blue or red, and chains classified as “noise” are colored white. (c) Number of droplets (n_d , upper) and sizes of droplets (s_d , lower) as functions of simulation time. In the lower plot, special markers are used for the two droplets in the high-density system (a) after 4.25×10^8 steps: the markers have yellow edges, with the face color being red for the larger droplet and blue for the smaller droplet, respectively. For the other dots and lines, yellow represents the high-density system, while green represents the low-density system.

On line 363, we have added this sentence:

In total, 1.225×10^9 and $\sim 1.200 \times 10^9$ steps were performed for the high- and low-density systems, respectively.

To avoid misunderstanding, we have changed all the “clusters” to “droplets” in the first paragraph on page 16. Accordingly, we have also changed “ n_c ” to “ n_d ” (number of droplets) and “ s_c ” to “ s_d ” (size of droplets) in the main text and Figure 4.

We have added the following paragraph on line 386:

These droplets further assembled into two after $\sim 3 \times 10^8$ steps. Eventually, after 8.5×10^8 steps, only one droplet remains, comprising $\sim 14,000$ chains and a diameter of approximately $0.1 \mu\text{m}$ (Fig. 4a and 4c). In contrast, the number of droplets in the low-density system decreases at a slower pace, reducing from $n_d = 53$ to $n_d = 16$ during the first 4×10^8 steps and eventually reaching 6 after 1.2×10^9 steps (Fig. 4c). Simultaneously, the largest droplet size slowly increases from $s_d = 452$ ($t = 0$) to $s_d = 1261$ at $t = 4 \times 10^8$ steps and finally to $s_d = 2,205$ at $t = 1.2 \times 10^9$ steps.

We have added the following paragraph on line 395:

Interestingly, we observed that the decrease in droplet number is not always due to the fusion of smaller droplets into larger ones, as shown in Fig. 3. We discovered that some droplets reduce in size and eventually dissolve into the dilute phase. For instance, in the high-density system, after 4.25×10^8 steps, the size (s_d) of the smaller droplet (depicted as blue in Fig. 4a, middle figure) decreased from 1,297 to 0 (as shown in Fig. 4c, bottom). Meanwhile, larger droplets can grow as the concentration in the dilute phase increases, with more chains entering the dense phase. For example, the size of the larger droplet in the high-density system increased from 12,850 at $t = 4.25 \times 10^8$ to 14,088 at $t = 1.225 \times 10^9$. The differing destinies of large and small droplets can be attributed to the pressure difference sustained across the interface between the two phases (Δp), which is described by the Young-Laplace equation, $\Delta p = -2\gamma/R_f$, where γ is the surface tension, and R_f is the mean curvature. For small, highly curved droplets, extra energy is required to overcome this large pressure, causing the diffusion of proteins into the dilute phase, and consequently driving Ostwald ripening.^{60,61} To the best of our knowledge, this is the first instance of using residue-resolution MD simulations to directly observe this phenomenon in a biomolecular condensation system.

Minor Comments

(8) The statement in line 188 a reference is missing.

We added the reference as 35 in the revised manuscript.

(9) The use of the term residue level CG models in the introduction is a bit misleading. It would seem more appropriate to split the mentioned CG models early on into those using

implicit and explicit solvent models in order to make clear that the DDS mostly focuses on the first class of models.

We added the term “implicit solvent” in the revised manuscript. Most correctly, our model can be explained as “implicit-solvent residue-level CG models”. We used the term in the revised manuscript (lines 62, 63, 65, 73, and so on).

Reviewer #2 (Remarks to the Author):

Jung and coworkers introduce an extension of the Genesis molecular dynamics engine that aims to solve the load-balancing problem, an issue that is prevalent when simulating systems with heterogeneous particle distributions, e.g., implicit-solvent coarse-grained molecular dynamics. The new code is referred to as CGDYN and the authors demonstrate its utility for large simulations of droplets undergoing fusion. The computational experiments are well-designed and the analysis sound. The manuscript is also well-written.

The main novelty presented by the manuscript is the cell-based kd-tree method that is used to perform heterogeneous domain decomposition with dynamic rebalancing during the simulation. However, based on my understanding of the presented methodology, the cell-based kd-tree method should yield comparable results to existing dynamic load balancing methods. For example, GROMACS does iterative dynamic load balancing where each cell is independently resized and staggered relative to other cells. LAMMPS has the recursive coordinate bisectioning method that is similar to the CGDYN approach. While perhaps not as widespread of a codebase, ls1 mardyn implements a kd-tree based dynamic load balancing approach that is perhaps most similar to the authors' approach. In other words, the heterogeneity of implicit-solvent coarse-grained molecular dynamics simulations can be largely mitigated by existing dynamic load balancing methods. Therefore, it is unclear to me how CGDYN improves upon the state-of-the-art other than incrementally.

Almost the same comment was given by the first reviewer, and we have already addressed this issue (our replies to the 3-th and 4-th comments by reviewer 1). Please see our replies and modifications in the revised manuscript.

(2) I am also surprised by the lack of comparison to existing load balancing methods. In the manuscript, the authors have only compared the performance of CGDYN to the group's other codes rather than any of the open-source MD engines mentioned in the introduction. While a quantitative comparison to other heterogeneous domain decomposition methods would be ideal, I understand that this may be a time-consuming process. At minimum, the authors should discuss CGDYN's approach in the context of the numerous load balancing methods that have been published that address the same

issue of particle heterogeneity. I mentioned some above and have compiled an incomplete list below:

This comment is essentially the same as the 6-th comment by the first reviewer.

As we discussed, we have added the performance comparisons with two more models/software, the dry Martini model using Gromacs and the HPS model using LAMMPS in this revision. In addition to the comparison with Open3SPN based on OpenMM, we carried out the comparison between CGDYN/GENESIS with three well-known MD software. In all the comparisons, we could show the superiority of the performance in CGDYN.

Besides, in the updated manuscript, we've expanded the length of our protein droplet dynamics simulations threefold, now encompassing approximately 1.2×10^9 steps for each system. This extension allowed us to capture fusion and dissolution events among the droplets. Notably, in the high-density system, we observed a transition to a single-droplet state. Interestingly, smaller droplets were seen to gradually shrink and ultimately dissolve into the dilute phase, whereas larger droplets exhibited growth (as depicted in the revised Figure 4). This dynamic closely mirrors the Ostwald ripening process typically seen in emulsions. To our knowledge, our study represents the first direct observation of such a phenomenon within biomolecular systems through residue-level CG MD simulations.

Slab simulations have been a common approach in computational studies of biomolecular condensation. These simulations employ a box with two shorter dimensions and one longer dimension, along with periodic boundary conditions, to quickly achieve equilibrium. However, this method does not accurately capture the influence of system size-dependent surface tension, which our simulations reveal to be critical in dictating the behavior of smaller droplets. This highlights the importance of utilizing our CGDYN software for more accurate droplet simulations in biomolecular condensation research.

Based on our results and discussions, it's crucial to highlight that the execution of such large-scale biological simulations featuring detailed residue-level descriptions of biomolecules would not be feasible without the technical advancements presented in this study. Consequently, we are confident that our work has forged a new pathway, bridging the gap between our comprehension of molecular structures and interactions and the broader field of sub-cellular dynamics. Please see more details from our reply to the 7th comment of the first reviewer.

Reviewer #3 (Remarks to the Author):

This paper presents a state-of-the-art implementation of simulation and modeling of coarse-grained macromolecular systems.

What is new:

(1) It is specifically designed to be efficient for systems with very large conformational changes. It focuses on efficiency for systems with unusual or weird particle distributions, as is often seen when implicit solvent is used.

What it is good for: Intrinsically disordered protein mixing. Very large mesoscopic simulations. One given example involves droplet formation, where homogeneous particles condense into a single dense droplet.

In the introduction, it should perhaps be noted that many CG simulation projects are also done using CHARMM. This is especially true for early CG simulations, and for those requiring Hessians. The list of seven programs seems incomplete with including CHARMM, for which there are many dozens of published CG projects.

In the original manuscript, we missed CHARMM although there have been many important developments of CG projects using CHARMM. In the introduction, we added the reference of CHARMM as a program name. We also added CG schemes with dynamic load balancing schemes in the revised manuscript.

In the revised manuscript on line 59, we modified the sentence:

Several MD programs, including CHARMM, Gromacs, OpenMM, NAMD, HOOMD-blue, LAMMPS, Cafemol, and GENESIS, offer a diverse set of tools and capabilities for performing CG MD simulations in various contexts.

(2) A new domain decomposition scheme with dynamic load balancing. The scalability of the new approach as shown by figure 2 is very good.

One outstanding questions relate to how well this approach will work on GPU based supercomputers. I believe the authors need to, at least, mention and address this issue and identify potential problems and difficulties that might arise for hardware that may become dominant in the future.

We are working on an all-atom MD speedup for GPU on a single node in a separate project. Similarly to the all-atom case, we think GPU-only usage on a single node is the optimal way for CG if the system size is not significantly large. However, when computing very large biomolecular systems that exceed the memory capacity of GPUs, such as a whole cell simulation, it may be reasonable to use a massively parallel supercomputer, whether it has GPUs or not. Even with GPU-based supercomputers, the main consideration is a good balance between communication and computation. In either case, the algorithm developed here is expected to be helpful as a fundamental technology for speeding up.

In the revised manuscript, on line 444, we added the sentences:

Our primary focus in this development is currently limited to CPU-based computers. Nonetheless, we anticipate that GPUs could further enhance the speed of MD simulations by employing the same domain decomposition with dynamic load balancing. While there have been several GPU implementations of CG potentials, optimizing multiple functions on GPU cores will pose significant challenges. In cases where the system size is not substantial, allocating all the energy/force calculations and integrations to GPUs may be feasible. However, when utilizing multiple nodes with GPUs, communication between different nodes becomes a more critical issue due to the reduced computation-to-communication ratio inherent in GPU usage.

Reviewers' Comments:

Reviewer #1:

Remarks to the Author:

I am satisfied with the changes to the manuscript. With the addition of new benchmarks comparing other MD engines, extended droplet simulations, and a discussion highlighting the advances of CGDYN more clearly the authors addressed my previous concerns. I still believe that the subject is somewhat too specialized for Nat. Commun., but that's a matter of personal preference. Thus I can recommend the publication of the manuscript.

Reviewer #2:

Remarks to the Author:

In this revised manuscript, Jung and coworkers have provided additional discussion and data that compares the CGDYN domain decomposition scheme to existing MD engines and have extended their MD simulations of liquid droplets. I appreciate the changes, which have addressed my major concerns, and especially appreciate the additional scaling calculations that compare CGDYN performance to that of GROMACS and LAMMPS across processor counts using existing CG models (dry Martini and HPS, respectively). I am now happy to recommend this manuscript for publication.

Reviewer #3:

Remarks to the Author:

The authors have responded well to all reviewers comments, and have made some significant and warranted improvements based on those. I have no additional issues to raise, and believe that the paper is publishable in its current form.

Reviewer #1 (Remarks to the Author):

I am satisfied with the changes to the manuscript. With the addition of new benchmarks comparing other MD engines, extended droplet simulations, and a discussion highlighting the advances of CGDYN more clearly the authors addressed my previous concerns. I still believe that the subject is somewhat too specialized for Nat. Commun., but that's a matter of personal preference. Thus I can recommend the publication of the manuscript.

Reviewer #1 (Remarks on code availability):

I have been able to find the source code as well as input files for all benchmarks using the URLs provided in the manuscript. Unfortunately, installing the code on MacOS for a quick check, was not possible. I'm sure by spending more time in resolving the dependency clashes it can be installed on MacOS as well.

We suspect you may have utilized LLVM C compiler on MacOS. Regrettably, GENESIS cannot be compiled using LLVM C. Instead, you can compile GENESIS by installing gcc compiler via Homebrew. We have added the guidelines for MacOS installation on GitHub for your reference.

Reviewer #2 (Remarks to the Author):

In this revised manuscript, Jung and coworkers have provided additional discussion and data that compares the CGDYN domain decomposition scheme to existing MD engines and have extended their MD simulations of liquid droplets. I appreciate the changes, which have addressed my major concerns, and especially appreciate the additional scaling calculations that compare CGDYN performance to that of GROMACS and LAMMPS across processor counts using existing CG models (dry Martini and HPS, respectively). I am now happy to recommend this manuscript for publication.

Reviewer #2 (Remarks on code availability):

Minor point - I only looked at the timing logs but noticed that some of the logs were incomplete (e.g., https://github.com/RikenSugitaLab/cgdyntest/blob/main/fugaku_droplet_benchmark/HPS_benchmark/lammps/D_p16_r01.log).

In the case of a small number of nodes, there were incomplete logs. This occurred due to assigning insufficient computation time in the job batch script. These incomplete logs were not utilized in our benchmark tests at all, and to prevent confusion, we have deleted them.

Reviewer #3 (Remarks to the Author):

The authors have responded well to all reviewers comments, and have made some significant and warranted improvements based on those. I have no additional issues to raise, and believe that the paper is publishable in its current form.